# Accuracy Analysis and Experimental Study of a Six-Dimensional Force Sensor with Circular Flexible Spherical Joints

**DOI:** 10.3390/s23167247

**Published:** 2023-08-18

**Authors:** Zhijun Wang, Xiaotao Zhang, Mengxiang Li

**Affiliations:** College of Mechanical Engineering, North China University of Science and Technology, Tangshan 063210, China; zjwang@ncst.edu.cn (Z.W.); mxli@stu.ncst.edu.cn (M.L.)

**Keywords:** circular flexible spherical joints, six-dimensional force sensor, stiffness matrix, force mapping matrix

## Abstract

This paper proposed a circular flexible spherical joint to reduce the processing difficulty and manufacturing cost, and design a six-dimensional force sensor based on it. This paper analyzes the influence of the structural parameters of the flexible spherical joints on the accuracy of the six-dimensional force sensor by comparing the force mapping matrix of flexible spherical and ideal spherical joints. As the force mapping matrix is related to the stiffness matrix, the stiffness matrix of a six-dimensional force sensor based on a circular flexible spherical joint and the force mapping matrix of the generalized external force acting on the sensor was derived. Finally, a set of optimal parameters was selected to create a prototype and was verified through finite elements and experiments. The results show that the six-dimensional force sensor with a circular flexible spherical joint has good accuracy and provides some guidance for the design and analysis of the sensor.

## 1. Introduction

Over the time, automation techniques have received extensive attention from many scholars. Artificial intelligence and robotics are important carriers of automated technologies, and six-dimensional force sensors, which are important force-sensing components, provide important tactile information for robots. According to the sensor measurement principle, six-dimensional force sensors can be divided into capacitive, piezoelectric, photoelectric and strain gauge types [1,2,3,4,5]. Strain gauge-type six-dimensional force sensors can be divided into integrated six-dimensional force sensors and parallel six-dimensional force sensors according to their elastomeric structure. There are various structural forms of integrated six-dimensional force sensors, the cross-beam structure is a representative structure of an integrated six-dimensional force transducer. Owing to the integral nature of its structure, the application of an external force in one direction will also cause changes in the strain gauge combination in other directions, resulting in a coupling error that cannot be completely decoupled [6,7,8]. In order to eliminate coupling errors in the six-dimensional force sensor, a parallel six-dimensional force sensor is created. The parallel six-dimensional force sensor adopts a ball joint connection method, with the force measuring branches being regarded as two-force rods subjected to axial forces only. According to the static analysis of the mechanism, when an external force is applied to the six-dimensional force sensor of the parallel structure, there is no stress coupling between the sensing elements of each measuring branch, and the structure is ingeniously decoupled [9,10].

Among the parallel six-dimensional force sensors, the Stewart platform is used for its high stiffness, high load-bearing capacity and non-accumulating position errors. In [11], it is recorded that Stewart from Germany designed the earliest six-dimensional force sensor with a spatial parallel structure. This type of sensor has the advantages of a compact structure and still has applications today. Because the six-dimensional force sensor connection described above uses a conventional spherical pair rather than an ideal spherical joint, the effect of gaps between the ball joints and frictional resistance cannot be completely eliminated. Meanwhile, the larger size of sensors manufactured using traditional spherical pairs is not conducive to the miniaturization development of six-dimensional force sensors. To address the aforementioned issues, experts and scholars have proposed corresponding improvement measures. In [12], traditional spherical pairs were replaced with conical ball–socket pairs, and preloading devices were added. The conical ball head reduces the contact area with the ball socket, and the pre- tightening device reduces the gap between the ball pairs, thereby improving the impact of friction/torque and gap on sensor accuracy. In the aforementioned six-dimensional force sensors, the over-sensor size problem caused by traditional ball-and-socket pairs has not been solved. Since flexible spherical joints do not have gaps and friction, which can improve the effects brought by traditional ball joints, some scholars have chosen to use flexible spherical joints for the development of six-dimensional force sensors. For example, the study in [13] designed a six-dimensional force sensor based on a Stewart platform. they replaced rotary joints with flexible spherical joints, and conducted finite element analysis. The results show that the rationally designed flexible spherical joint can replace the traditional ball joint. The study in [14] proposes a novel parallel six-dimensional force sensor with horizontal and vertical distribution of measurement branches and flexible ball hinges in the connecting part, and its hydrostatic model shows that this structure effectively improves the decoupling performance of the sensor. Currently, most flexible spherical joints used in sensor applications are cylindrical flexible spherical joints, which have disadvantages such as high processing difficulty and difficulty in installation difficulty. To solve this problem, a circular flexible joint is proposed and installed on a six-dimensional force sensor. Compared to traditional flexible joints, circular flexible joints have the advantages of simple processing, easy installation, and reduced manufacturing costs. Meanwhile, flexible spherical joints cannot be fully equated with ideal ball hinges, which can cause coupling errors in six-dimensional force sensors, thus reducing the accuracy of the sensors. Therefore, it is necessary to consider the impact of circular flexible spherical joints on the measurement accuracy of the sensor. To reduce the coupling error and improve precision, it is necessary to design the parameters of the flexible spherical joint. The accuracy of the six-dimensional force sensor is mainly related to the force mapping matrix of the sensor. Therefore, this paper introduces the concept of Euclidean distance to analyze the force mapping matrices of the flexible spherical joint and the ideal spherical hinge, and the closer the force mapping matrices of the flexible spherical joint and the ideal spherical hinge are, the higher the accuracy of the sensor is.

Since it is not possible to directly approximate the degree of the bipartite matrix, an indirect method is used in this study. First, we introduce the concept of Euclidean, and the similarity of the force mapping matrix was determined by comparing the Euclidean distance of the force mapping matrix between the flexible spherical joint and the ideal spherical joint six-dimensional force sensor. secondly, using this method, we analyze the influence of the structural parameters of the flexible joint on the measurement accuracy of the sensor was analyzed. Finally, a set of optimal parameters are selected to make a prototype and verified by finite element and experiment.

## 2. Configuration and Stiffness Modeling of the Novel Circular Flexible Spherical Joints

### 2.1. Configuration of the Novel Circular Flexible Spherical Joint

Flexible spherical joints are components that weaken their local stiffness through notches, and undergo bending deformation under the action of torque. Bending deformation mainly occurs at the weakest notch [15]. The proposed structure of a circular flexible spherical joint proposed based on the definition of a flexible spherical joint is shown in Figure 1.

In Figure 1, the circular through hole in the middle is the connecting part, which can be connected to the circular flexible spherical joint with bolts. O is the center of the circular through-hole on the upper surface. $O_{1}$ is the center of a circular notch. The distance from the center of the circle O to the outer circle of the circular notch is r. The radius of the circular notch is R and the distance between its two endpoints is l. The upper and lower circular notches were symmetrically distributed at the center of the flexible spherical joint. The distance between the weakest points of the upper and lower circular notch is t. A flexible spherical joint has good applicability in many scenarios and has the following advantages.

Easy to process, circular notch can be milled to reduce processing difficulty.Strong applicability, suitable for various occasionsIt has good isotropic performance and adopts a symmetrical structure, achieving the same stiffness in both directions.

### 2.2. Stiffness Modeling of the Novel Circular Flexible Spherical Joints

A circular flexible spherical joint is shown in Figure 1. To model its stiffness, the flexible spherical joint was first divided into several parts using the segmentation method. The segmented figure can be regarded as a compliant mechanism, and its specific form is illustrated in Figure 2.

First, the stiffness matrix of the compliant mechanism is calculated using a numerical analysis method [16]. Secondly, coordinate transformation was performed using the pose transformation matrix. Finally, because compliant mechanisms have the same reference coordinate system, the stiffness matrix of the overall circular flexible spherical joint can be obtained by adding the stiffness matrices of the same reference coordinate system. The stress of compliant mechanism is mainly concentrated in the groove, the deformation caused by the stress is mainly concentrated in the groove, and the rest can be regarded as a rigid body. From this, an approximate equation for the deflection curve of the beam-bending deformation can be obtained,
(1)$$\Delta_{z}=\frac{d^{2}\Delta_{z}}{dz^{2}}=\frac{M\left(y\right)}{EI\left(y\right)}$$

Here, *E* donates elastic modulus.
$$\begin{array}{l} M\left(y\right)=M_{x}+F_{z}\left(l-y\right)\\ I\left(y\right)=b{\left\{t+2\left[R-\sqrt{R^{2}-{\left(\frac{l}{2}-y\right)}^{2}}\right]\right\}}^{3}/12\\ y\in \left[0,l\right] \end{array}$$

The calculation and solution of this differential equation is very complex, and approximate solutions can be obtained using numerical analysis method.

Let $\theta \left(y\right)={\Delta^{\prime }}_{z}=\frac{d\Delta_{z}}{dy}$, then Equation (1) becomes $\theta^{\prime }\left(y\right)=f\left(y\right)=\frac{M\left(y\right)}{EI\left(y\right)}$.

The Runge–Kutta formula is an important numerical solution method which has a wide range of applications in the engineering field, with the advantages of high accuracy, good stability and wide applicability. In this paper, we use the second-order Runge–Kutta formula to solve, and the second-order Runge–Kutta formula is as follows,
(2)$$\begin{array}{l} \theta_{i+1}=\theta_{i}+\frac{h}{2}\left(k_{1}+k_{2}\right)\\ k_{1}=f(x_{i})\\ k_{2}=f\left(x_{i}+h\right) \end{array}\;(i=0,1,2\cdots \cdots n-1)$$

Here, n=4. h=l/4.

Let n=4, h=l/4, calculated by the second-order Runge–Kutta formula, the corresponding relationship between $x_{i}$ and $\theta_{i}$ is shown in Table 1.

Here, $h_{1}=t+2R-\sqrt{4R^{2}-l^{2}}$, $h_{2}=t+2R-\sqrt{16R^{2}-l^{2}}/2$.

The values in Table 1 can be approximated by Lagrange Polynomial Interpolation to obtain θ(y). θ(y) is the angle of rotation around x can be obtained,
(3)$$\theta \left(y\right)=\sum_{i=0}^{4}\left(\sum_{\begin{array}{l} i=0\\ i\neq j \end{array}}^{4}\frac{y-y_{j}}{y_{i}-y_{j}}\right)\theta_{i}$$

The equation at the end is $\theta \left(l\right)=\frac{3l\left(F_{z}l+2M_{x}\right)}{2Eb}\left(\frac{1}{h_{1}^{3}}+\frac{2}{h_{2}^{3}}+\frac{1}{t^{3}}\right)$.

Equation (3) is the expression of the rotation angle of the compliant mechanism around the x-axis, that is, the expression of the end. The end expression is can be obtained,
(4)$$\theta_{x}=\frac{3l\left(F_{z}l+2M_{x}\right)}{2Eb}\left(\frac{1}{h_{1}^{3}}+\frac{2}{h_{2}^{3}}+\frac{1}{t^{3}}\right)$$

Using the Runge–Kutta formula again for ${\Delta^{\prime }}_{z}=\theta \left(x\right)$, the corresponding relationship between *y_i_* and $\Delta_{i}$ is obtained, as shown in Table 2.

Perform Lagrange Polynomial Interpolation on Table 2 to obtain the displacement expression at the end along the *z*-axis can be obtained,
(5)$$\Delta_{z}=\frac{3l^{2}}{16Eb}\left(\frac{7F_{z}l+8M_{x}}{h_{1}^{3}}+\frac{10F_{z}l+16M_{x}}{h_{2}^{3}}+\frac{4F_{z}l+8M_{x}}{t^{3}}\right)$$

By combining Equations (4) and (5), the relationship between *F_z_*, $M_{z}$, and $\theta_{x}$, $\Delta_{z}$ can be obtained as follows,
(6)$$F_{z}=\frac{16Ebh_{1}^{3}h_{2}^{3}}{3l^{3}\left(3h_{2}^{3}+2h_{1}^{3}\right)}\Delta_{z}-\frac{8Ebh_{1}^{3}h_{2}^{3}}{3l^{2}\left(3h_{2}^{3}+2h_{1}^{3}\right)}\theta_{x}$$
(7)$$M_{x}=\frac{-8Ebh_{1}^{3}h_{2}^{3}}{3l^{2}\left(3h_{2}^{3}+2h_{1}^{3}\right)}\Delta_{z}+\frac{Eb\left(4h_{1}^{3}h_{2}^{3}+7t^{3}h_{2}^{3}+10t^{3}h_{1}^{3}\right)h_{1}^{3}h_{2}^{3}}{3l\left(h_{1}^{3}h_{2}^{3}+t^{3}h_{2}^{3}+t^{3}h_{1}^{3}\right)\left(3h_{2}^{3}+2h_{1}^{3}\right)}\theta_{x}$$

Similarly, the relationship between $F_{x}$, $M_{z}$, and $\theta_{z}$, $\Delta_{x}$ can be obtained as follows,
(8)$$F_{x}=-\frac{8Eb^{3}h_{1}h_{2}}{3l^{2}\left(3h_{2}+2h_{1}\right)}\theta_{z}+\frac{16Eb^{3}h_{1}h_{2}}{3l^{3}\left(3h_{2}+2h_{1}\right)}\Delta_{x}$$
(9)$$M_{z}=\frac{Eb^{3}h_{1}h_{2}}{3l}\left[\frac{4h_{1}h_{2}+10h_{1}t+7h_{2}t}{\left(2h_{1}t+h_{2}t+h_{1}h_{2}\right)\left(3h_{2}+2h_{1}\right)}\right]\theta_{z}-\frac{8Eb^{3}h_{1}h_{2}}{3l^{2}\left(3h_{2}+2h_{1}\right)}\Delta_{x}$$

The axial tensile deformation of the compliant mechanism under the action of $F_{y}$ can be obtained as follows,
(10)$$\partial_{y}=\int_{l}\frac{F\left(y\right)}{EA\left(y\right)}dy=\int_{0}^{l}\frac{F_{y}dy}{Eb\left[t+2R-\sqrt{4R^{2}-{\left(l-2y\right)}^{2}}\right]}$$

The integral is difficult to find the result directly, and the complex parabolic product formula is a kind of complex product formula, with the advantages of high accuracy, low operation and numerical stability. This example can be solved by the complex parabolic product formula, the complex parabolic product formula is as follows,
(11)$$S_{2m}=\frac{h}{3}\left[f\left(a\right)+f\left(b\right)+2\sum_{k=1}^{m-1}f\left(x_{2k}\right)+4\sum_{k=0}^{m-1}f\left(x_{2k-1}\right)\right]$$

Here, [*a*, *b*] is the integral interval. Divide the interval [*a*, *b*] into 2 m equal parts. h=(b−a)/2. Node is $x_{k}=a+kh$ (k=0,1,2⋯2m), In this article, m is taken as 2.

After the above analysis, it can be concluded that $\Delta_{y}$ can be obtained as follows,
(12)$$\Delta_{y}=\frac{l}{6Eb}\left(\frac{1}{h_{1}}+\frac{1}{t}+\frac{4}{h_{2}}\right)F_{y}$$

It can be concluded that $F_{y}$ can be obtained as follows,
(13)$$F_{y}=\frac{6Ebth_{1}h_{2}}{l\left(th_{2}+h_{1}h_{2}+4th_{1}\right)}\Delta_{y}$$

The axial torsional deformation of the compliant mechanism under the action of $M_{y}$ can be obtained as follows,
(14)$$\theta_{y}=\int_{L}\frac{3M_{y}dy}{b^{3}\left[t+2R-\sqrt{4R^{2}-{\left(l-2y\right)}^{2}}\right]}$$

This integral is difficult to find the result directly and can be obtained by compounding the parabolic product formula. Similarly, in this example, let m=2. The complexed parabolic product formula is solved as shown below,
(15)$$\theta_{y}=\frac{l}{2b^{3}}\left(\frac{1}{h_{1}}+\frac{1}{t}+\frac{4}{h_{2}}\right)M_{y}$$

It can be concluded that $M_{y}$ can be obtained as follows,
(16)$$M_{y}=\frac{2Gb^{3}th_{1}h_{2}}{l\left(th_{2}+h_{1}h_{2}+th_{1}\right)}\theta_{y}$$

Here, G=E/2(1+μ).

By using the corresponding relationship obtained above, the overall corresponding relationship can be obtained,
(17)$$F=K_{1}X$$

Here, $F={\left[\begin{array}{cccccc} F_{x} & F_{y} & F_{z} & M_{x} & M_{y} & M_{z} \end{array}\right]}^{T}$. $X={\left[\begin{array}{cccccc} \Delta_{x} & \Delta_{y} & \Delta_{z} & \theta_{x} & \theta_{y} & \theta_{z} \end{array}\right]}^{T}$.

$K_{1}$ is the stiffness matrix of compliant mechanism.
(18)$$K_{1}=\left[\begin{array}{cccccc} A_{1} & 0 & 0 & 0 & 0 & B_{1}\\ 0 & C_{1} & 0 & 0 & 0 & 0\\ 0 & 0 & D_{1} & E_{1} & 0 & 0\\ 0 & 0 & E_{1} & F_{1} & 0 & 0\\ 0 & 0 & 0 & 0 & G_{1} & 0\\ B_{1} & 0 & 0 & 0 & 0 & H_{1} \end{array}\right]$$
where $A_{1}=\frac{16Eb^{3}h_{1}h_{2}}{3l^{3}\left(3h_{2}\;+\;2h_{1}\right)}$, $B_{1}=-\frac{8Eb^{3}h_{1}h_{2}}{3l^{2}\left(3h_{2}\;+\;2h_{1}\right)}$, $C_{1}=\frac{6Ebth_{1}h_{2}}{l\left(th_{2}\;+\;h_{1}h_{2}\;+\;4th_{1}\right)}$, $D_{1}=\frac{16Ebh_{1}^{3}h_{2}^{3}}{3l^{3}\left(3h_{2}^{3}\;+\;2h_{1}^{3}\right)}$ $E_{1}=-\frac{8Ebh_{1}^{3}h_{2}^{3}}{3l^{2}\left(3h_{2}^{3}\;+\;2h_{1}^{3}\right)}$, $F_{1}=\frac{Eb\left(4h_{1}^{3}h_{2}^{3}\;+\;7t^{3}h_{2}^{3}\;+\;10t^{3}h_{1}^{3}\right)h_{1}^{3}h_{2}^{3}}{3l\left(h_{1}^{3}h_{2}^{3}\;+\;t^{3}h_{2}^{3}\;+\;t^{3}h_{1}^{3}\right)\left(3h_{2}^{3}\;+\;2h_{1}^{3}\right)}$, $G_{1}=\frac{2G_{1}b^{3}th_{1}h_{2}}{l\left(th_{2}\;+\;h_{1}h_{2}\;+\;4th_{1}\right)}$, $H_{1}=\frac{Eb^{3}h_{1}h_{2}}{3l}\left[\frac{4h_{1}h_{2}\;+\;10h_{1}t\;+\;7h_{2}t}{\left(h_{1}t\;+\;2h_{2}t\;+\;h_{1}h_{2}\right)\left(3h_{2}\;+\;2h_{1}\right)}\right]$.

A circular flexible spherical joint is composed of multiple flexible structures as shown in Figure 2. The reference coordinate system is located at the geometric center of the upper surface of the flexible spherical joint, and the angle of rotation of the reference coordinate system of the compliant mechanism around the z-axis relative to the fixed coordinate system at any position is set as γ. Before the stiffness matrix of the circular flexible spherical joint can be calculated, the stiffness matrix of the flexible mechanism needs to be mapped to the same reference coordinate system, therefore, the coordinate mapping equation for the flexibility matrix must first be established. According to the literature [17,18], the mapping equation for the stiffness matrix of the flexible mechanism from coordinate system *i* to coordinate system *j* is as follows, with the reference coordinate system located at the geometric center of the upper surface of the flexible spherical joint.
(19)$$K_{j}=T_{i}^{j}K_{i}{\left(T_{i}^{j}\right)}^{T}$$
(20)$$T_{i}^{j}=\left[\begin{array}{cc} R_{i}^{j} & S\left(r_{i}^{j}\right)R_{i}^{j}\\ 0 & R_{i}^{j} \end{array}\right]$$

Here, *K_i_* represents the stiffness matrix in the reference coordinate system i, $T_{i}^{j}$ represents the transformation matrix from reference coordinate system i to coordinate system j. $R_{i}^{j}$ represents the rotation matrix from reference coordinate system i to coordinate system j. $r_{i}^{j}$ represents the position coordinate of reference coordinate system i to coordinate system. $S\left(r_{i}^{j}\right)$ represents a diagonal symmetric matrix of positional coordinates.

Substituting the values of the coordinate system into the formula yields the following expression.
(21)ri2=(rcosγrsinγ−m2)T
(22)S(ri2)=[0m2rsinγ−m20−rcosγ−rsinγrcosγ0]

Here, r is the radius of the outer circle of the groove. m is the overall thickness.
(23)Ri2=[cos(−γ)−sin(−γ)0sin(−γ)cos(−γ)0001]=[cosγsinγ0−sinγcosγ0001]
(24)Ti2=[cosγsinγ0−m2sinγm2cosγrsinγ−sinγcosγ0−m2cosγ−m2sinγ−rcosγ001−rsin2γrcos2γ0000cosγsinγ0000−sinγcosγ0000001]
(25)(Ti2)T=[cosγ−sinγ0000sinγcosγ0000001000−m2sinγ−m2cosγ−rsin2γcosγ−sinγ0m2cosγ−m2sinγrcos2γsinγcosγ0rsinγ−rcosγ0001]

According to the pose transformation formula, the stiffness matrix $K_{2i}$ of any compliant mechanism can be obtained,
(26)K2i=Ti2K1(Ti2)T

The flexible spherical joint was divided into 720 parts, each of which can be approximately regarded as a compliant mechanism as shown in Figure 2. The side length of the compliant mechanism is b=2πr/720. As each compliant mechanism has the same reference coordinate system, the overall stiffness matrix of the circular flexible spherical joint is equal to the sum of the stiffness matrices of each compliant mechanism. So,
(27)$$K_{2}=\sum_{i=1}^{720}K_{2i}$$

Owing to the order-of-magnitude differences, discarding a portion of the values will not have a significant impact on the final result. A larger coefficient for the stiffness coefficient of movement in the direction of x,y,z, and a smaller coefficient for the stiffness coefficient of rotation in the direction of x,y,z. When a corresponding force or torque is applied, the main change occurs as rotation or movement in the corresponding direction. Therefore, only the values of the main diagonal elements of the stiffness matrix were retained, and the final determination of the stiffness matrix $K_{2}$ was as follows,
(28)$$K_{2}=\left[\begin{array}{cccccc} 360C_{1} & 0 & 0 & 0 & 0 & 0\\ 0 & 360C_{1} & 0 & 0 & 0 & 0\\ 0 & 0 & 720D_{1} & 0 & 0 & 0\\ 0 & 0 & 0 & 360G_{1} & 0 & 0\\ 0 & 0 & 0 & 0 & 360G_{1} & 0\\ 0 & 0 & 0 & 0 & 0 & 720H_{1} \end{array}\right]$$

## 3. Configuration Six-Dimensional Force Sensor with Circular Flexible Spherical Joints

A schematic of the structure of the six-dimensional force sensor designed according to the new annular flexible ball hinge is shown in Figure 3. The six-dimensional force sensor consists of an upper platform, a lower platform, and six SPS force-measuring branches. o−xyz is the Cartesian coordinate system for the center of the upper platform. The z-axis is perpendicular to the upper platform and vertically upward. The three vertical branches are evenly distributed along the z-axis into an equilateral triangle, the center of the circumscribed circle is o and the radius is $r_{1}$. $A_{i(i=1,2,3,4,5,6)}$ is the ideal spherical joint on the lower platform. $a_{i(i=1,2,3,4,5,6)}$ is the ideal spherical joint on the upper platform. $a_{1}$ is on the x-axis of the o−xyz coordinate system. The plane formed by the three horizontal branches is parallel to and between the upper and lower platforms. The vertical branches of $A_{i(i=4,5,6)}$ are evenly distributed along the z-axis and on a circle with a radius of $r_{2}$, and the vertical branches of $a_{i(i=1,2,3)}$ are evenly distributed along the z-axis and on a circle with a radius of $r_{3}$. $o^{\prime }$ is on the z-axis of the o−xyz coordinate system. The angle between the $o^{\prime }A_{4}$ and the planes of xoz is α. The angle between the $o^{\prime }a_{4}$ and the planes of xoz is β. The distance from the plane formed by the horizontal branches of the plane to the detection platform is $l_{1}$. The distance between the upper and lower platforms is l. F and M are the external forces and moments acting on the upper platform.

According to the above description, the coordinates of ideal spherical joints $A_{i(i=1,2,3,4,5,6)}$ and $a_{i(i=1,2,3,4,5,6)}$ in the Cartesian coordinate system of o−xyz can be obtained as follows,
(29)$$A={\left[\begin{array}{c} A_{1}\\ A_{2}\\ A_{3}\\ A_{4}\\ A_{5}\\ A_{6} \end{array}\right]}^{T}=\left[\begin{array}{cccccc} r_{1} & r_{1}\cos\frac{2\pi }{3} & r_{1}\cos\frac{4\pi }{3} & r_{2}\cos\alpha & r_{2}\cos\left(\alpha +\frac{2\pi }{3}\right) & r_{2}\cos\left(\alpha +\frac{4\pi }{3}\right)\\ 0 & r_{1}\sin\frac{2\pi }{3} & r_{1}\sin\frac{4\pi }{3} & r_{2}\sin\alpha & r_{2}\sin\left(\alpha +\frac{2\pi }{3}\right) & r_{2}\sin\left(\alpha +\frac{4\pi }{3}\right)\\ -l & -l & -l & -l_{1} & -l_{1} & -l_{1} \end{array}\right]$$
(30)$$a={\left[\begin{array}{c} a_{1}\\ a_{2}\\ a_{3}\\ a_{4}\\ a_{5}\\ a_{6} \end{array}\right]}^{T}=\left[\begin{array}{cccccc} r_{1} & r_{1}\cos\frac{2\pi }{3} & r_{1}\cos\frac{4\pi }{3} & r_{3}\cos\beta & r_{3}\cos\left(\beta +\frac{2\pi }{3}\right) & r_{3}\cos\left(\beta +\frac{4\pi }{3}\right)\\ 0 & r_{1}\sin\frac{2\pi }{3} & r_{1}\sin\frac{4\pi }{3} & r_{3}\sin\beta & r_{3}\sin\left(\beta +\frac{2\pi }{3}\right) & r_{3}\sin\left(\beta +\frac{4\pi }{3}\right)\\ 0 & 0 & 0 & -l_{1} & -l_{1} & -l_{1} \end{array}\right]$$

When external forces and moments act on the upper platform, according to the static equilibrium equation, it can be inferred as follows,
(31)$$\left[\begin{array}{c} F\\ M \end{array}\right]=Gf$$

Here, $F={\left[\begin{array}{ccc} F_{x} & F_{y} & F_{z} \end{array}\right]}^{T}$ a is the external force acting on the upper platform. $M={\left[\begin{array}{ccc} M_{x} & M_{y} & M_{z} \end{array}\right]}^{T}$ is the external moment acting on the upper platform. $f={\left[\begin{array}{cccccc} f_{1} & f_{2} & f_{3} & f_{4} & f_{5} & f_{6} \end{array}\right]}^{T}$ is the axial force acting on each force-measuring branch. G is the force mapping matrix. Its specific form can be expressed according to the spiral theory, and the specific form is as follows,
(32)$$G=\left[\begin{array}{cccccc} e_{1}^{O} & e_{2}^{O} & e_{3}^{O} & e_{4}^{O} & e_{5}^{O} & e_{6}^{O}\\ a_{1}^{O}\times e_{1}^{O} & a_{2}^{O}\times e_{2}^{O} & a_{3}^{O}\times e_{3}^{O} & a_{4}^{O}\times e_{4}^{O} & a_{5}^{O}\times e_{5}^{O} & a_{6}^{O}\times e_{6}^{O} \end{array}\right]$$

$e_{i(i=1,2\ldots 6)}^{O}=\left(a_{i}^{O}-A_{i}^{O}\right)/\left|a_{i}^{O}-A_{i}^{O}\right|$ is the unit vector of each measurement branch in the o−xyz coordinate system.

It is calculated that,
(33)$$G=\left[\begin{array}{cccccc} 0 & 0 & 0 & \frac{r_{3}\cos\beta \;-\;r_{2}\cos\alpha }{\sqrt{r_{2}^{2}\;+\;r_{3}^{2}\;-\;2r_{3}^{2}\cos\left(\beta \;-\;\alpha \right)}} & \frac{r_{3}\cos\left(\beta \;+\;\frac{2\pi }{3}\right)\;-\;r_{2}\cos\left(\alpha \;+\;\frac{2\pi }{3}\right)}{\sqrt{r_{2}^{2}\;+\;r_{3}^{2}\;-\;2r_{3}^{2}\cos\left(\beta \;-\;\alpha \right)}} & \frac{r_{3}\cos\left(\beta \;+\;\frac{4\pi }{3}\right)\;-\;r_{2}\cos\left(\alpha \;+\;\frac{4\pi }{3}\right)}{\sqrt{r_{2}^{2}\;+\;r_{3}^{2}\;-\;2r_{3}^{2}\cos\left(\beta \;-\;\alpha \right)}}\\ 0 & 0 & 0 & \frac{r_{3}\sin\beta \;-\;r_{2}\sin\alpha }{\sqrt{r_{2}^{2}\;+\;r_{3}^{2}\;-\;2r_{3}^{2}\cos\left(\beta \;-\;\alpha \right)}} & \frac{r_{3}\sin\left(\beta \;+\;\frac{2\pi }{3}\right)\;-\;r_{2}\sin\left(\alpha \;+\;\frac{2\pi }{3}\right)}{\sqrt{r_{2}^{2}\;+\;r_{3}^{2}\;-\;2r_{3}^{2}\cos\left(\beta \;-\;\alpha \right)}} & \frac{r_{3}\sin\left(\beta \;+\;\frac{4\pi }{3}\right)\;-\;r_{2}\sin\left(\alpha \;+\;\frac{4\pi }{3}\right)}{\sqrt{r_{2}^{2}\;+\;r_{3}^{2}\;-\;2r_{3}^{2}\cos\left(\beta \;-\;\alpha \right)}}\\ 1 & 1 & 1 & 0 & 0 & 0\\ 0 & \frac{\sqrt{3}r_{1}}{2} & \frac{\sqrt{3}r_{1}}{-2} & \frac{l_{1}r_{3}\sin\beta \;-\;l_{1}r_{2}\sin\alpha }{\sqrt{r_{2}^{2}\;+\;r_{3}^{2}\;-\;2r_{3}^{2}\cos\left(\beta \;-\;\alpha \right)}} & \frac{l_{1}r_{3}\sin\left(\beta \;+\;\frac{2\pi }{3}\right)\;-\;l_{1}r_{2}\sin\left(\alpha \;+\;\frac{2\pi }{3}\right)}{\sqrt{r_{2}^{2}\;+\;r_{3}^{2}\;-\;2r_{3}^{2}\cos\left(\beta \;-\;\alpha \right)}} & \frac{l_{1}r_{3}\sin\left(\beta \;+\;\frac{4\pi }{3}\right)\;-\;l_{1}r_{2}\sin\left(\alpha \;+\;\frac{4\pi }{3}\right)}{\sqrt{r_{2}^{2}\;+\;r_{3}^{2}\;-\;2r_{3}^{2}\cos\left(\beta \;-\;\alpha \right)}}\\ r_{1} & \frac{r_{1}}{2} & \frac{r_{1}}{2} & \frac{l_{1}r_{2}\cos\alpha \;-\;l_{1}r_{3}\cos\beta }{\sqrt{r_{2}^{2}\;+\;r_{3}^{2}\;-\;2r_{3}^{2}\cos\left(\beta \;-\;\alpha \right)}} & \frac{l_{1}r_{2}\cos\left(\alpha \;+\;\frac{2\pi }{3}\right)\;-\;l_{1}r_{3}\cos\left(\beta \;+\;\frac{2\pi }{3}\right)}{\sqrt{r_{2}^{2}\;+\;r_{3}^{2}\;-\;2r_{3}^{2}\cos\left(\beta \;-\;\alpha \right)}} & \frac{l_{1}r_{2}\cos\left(\alpha \;+\;\frac{4\pi }{3}\right)\;-\;l_{1}r_{3}\cos\left(\beta \;+\;\frac{4\pi }{3}\right)}{\sqrt{r_{2}^{2}\;+\;r_{3}^{2}\;-\;2r_{3}^{2}\cos\left(\beta \;-\;\alpha \right)}}\\ 0 & 0 & 0 & \frac{r_{2}r_{3}\sin\left(\beta \;-\;\alpha \right)}{\sqrt{r_{2}^{2}\;+\;r_{3}^{2}\;-\;2r_{3}^{2}\cos\left(\beta \;-\;\alpha \right)}} & \frac{r_{2}r_{3}\sin\left(\beta \;-\;\alpha \right)}{\sqrt{r_{2}^{2}\;+\;r_{3}^{2}\;-\;2r_{3}^{2}\cos\left(\beta \;-\;\alpha \right)}} & \frac{r_{2}r_{3}\sin\left(\beta \;-\;\alpha \right)}{\sqrt{r_{2}^{2}\;+\;r_{3}^{2}\;-\;2r_{3}^{2}\cos\left(\beta \;-\;\alpha \right)}} \end{array}\right]$$

From Equations (31) and (33), it can be seen that the six-dimensional force sensor with parallel structure can complete partial decoupling, $F_{x}$ and $F_{y}$ are only related to $f_{4}$, $f_{5}$ and $f_{6}$, and $F_{z}$ is only related to $f_{1}$, $f_{2}$ and $f_{3}$. According to Figure 3, $f_{1}$, $f_{2}$ and $f_{3}$ is the axial force measured by the vertical force measuring branch. $f_{4}$, $f_{5}$ and $f_{6}$ is the axial force measured by the vertical force measuring branch. The partially decoupled six-dimensional force transducer can reduce the measurement error of the sensor.

According to the schematic diagram of the six-dimensional force sensor structure, the novel circular flexible spherical joints six-dimensional force sensor designed is shown in Figure 4. The sensor consists of an upper platform, a lower platform, six circular flexible ball joints, six columns, and six measurement branches.

## 4. Stiffness Modeling of the Six-Dimensional Force Sensor with Circular Flexible Spherical Joints

As the generalized external force applied to the sensor and the force mapping matrix of the measurement branch are related to the stiffness matrix of the sensor, it is necessary to establish the stiffness matrix of the sensor [19]. In this paper, we use the theory that the stiffness of a parallel mechanism is equal to the sum of the stiffness matrices of each branch and that stiffness and flexibility are inverse matrices of each other. First, the sensor system is decomposed into basic elements, and the stiffness matrix of the basic elements is established, which is then these stiffness matrices are used to establish the overall stiffness matrix of the sensor. Figure 5 shows the decomposed diagram of the measuring branch, where part 1 and part 3 are circular flexible spherical joints with the same structure, and the part 2 is a flexible beam with a rectangular cross-section, and the surface of the beam is pasted with strain gauges. Therefore, it is necessary to establish the stiffness matrix of a circular flexible spherical joint and a flexible beam with a rectangular cross-section.

### 4.1. Stiffness Matrix Modeling of Flexible Beam with a Rectangular Cross-Section

The deformation of the free end of the cantilever beam when subjected to a generalized external force is shown in Figure 6. $O_{g}X_{g}Y_{g}Z_{g}$ is a fixed coordinate system. $O_{p}X_{p}Y_{p}Z_{p}$ is the reference coordinate system, ${O^{\prime }}_{p}{X^{\prime }}_{p}{Y^{\prime }}_{p}{Z^{\prime }}_{p}$ is the attitude coordinate system under the generalized external force. $F_{x}$, $F_{y}$, $F_{z}$, $M_{x}$, $M_{y}$, $M_{z}$ are the external force and moment acting on the free end of the cantilever beam. The values of $Z_{p}$ and ${Z^{\prime }}_{p}$ are upward along the direction of the cantilever beam, and the directions of the other axes are distributed according to the right-hand rule. Due to the need to consider the size and range of the six-dimension force sensor, the aspect ratio of the flexible beam is usually small, so the shear effect of the beam should be considered when calculating the stiffness matrix. According to Timoshenko beam theory [20,21,22], the stiffness matrix $K_{f}$ of square section beam can be expressed as follows,
(34)$$C_{f}=\left[\begin{array}{cccccc} \frac{4l^{3}}{Ea^{4}}+\frac{6l}{5Ga^{2}} & 0 & 0 & 0 & \frac{6l^{2}}{Ea^{4}} & 0\\ 0 & \frac{4l^{3}}{Ea^{4}}+\frac{6l}{5Ga^{2}} & 0 & -\frac{6l^{2}}{Ea^{4}} & 0 & 0\\ 0 & 0 & \frac{l}{Ea^{2}} & 0 & 0 & 0\\ 0 & -\frac{6l^{2}}{Ea^{4}} & 0 & \frac{12l}{Ea^{4}} & 0 & 0\\ \frac{6l^{2}}{Ea^{4}} & 0 & 0 & 0 & \frac{12l}{Ea^{4}} & 0\\ 0 & 0 & 0 & 0 & 0 & \frac{64l}{9Ga^{4}} \end{array}\right]$$
(35)$$K_{f}=C_{f}^{-1}$$

Here, $K_{f}$ is the stiffness matrix of the cantilever beam. $C_{f}$ is the flexibility matrix of the cantilever beam. *G* denotes shear modulus. *l* denotes the length of the beam.

### 4.2. Stiffness Matrix Modeling of Measurement Branch

As we all know, to calculate the overall stiffness matrix of the test branch, we first need to map the coordinate system of each part to the same reference coordinate system according to Equations (19) and (20), and then add the stiffness matrix of each part [23,24].

The force-sensitive element is broken down into the first, second, and third parts, and since the stiffness matrices of the first and third parts are the same, only the part 1 and 3 stiffness matrices need to be calculated. This is shown in Figure 7.

$O_{i}-X_{i}Y_{i}Z_{i}$ is the fixed coordinate system at the bottom center of the Part 1, i=1,2,3,4,5,6 is the serial number of the force measurement branch, $O_{i1}-X_{i1}Y_{i1}Z_{i1}$ is the reference coordinate system acting at the end center of the part 1, $O_{i2}-X_{i2}Y_{i2}Z_{i2}$ is the reference coordinate system acting on the end center of the part 2, $O_{i3}-X_{i3}Y_{i3}Z_{i3}$ is the reference coordinate system acting on the end center of the part 3, $O_{g1}-X_{g1}Y_{g1}Z_{g1}$ is the reference coordinate system that coincides with $O_{i3}-X_{i3}Y_{i3}Z_{i3}$. The Z-axis of the above coordinate system is collinear, and the remaining axes are determined according to the right-hand rule, and the corresponding axes are parallel to each other.

According to the virtual work principle and the deformation Superposition principle, the stiffness matrix of the measured branch is as follows,
(36)$$K_{i}^{gi}=\sum_{j=1}^{3}T_{ij}^{gi}K_{fj}{\left(T_{ij}^{gi}\right)}^{T}$$

Here, $T_{ij}^{gi}=\left[\begin{array}{cc} R_{ij}^{gi} & S\left(r_{ij}^{gi}\right)R_{ij}^{gi}\\ 0 & R_{ij}^{gi} \end{array}\right]$. $r_{i1}^{gi}={\left(\begin{array}{ccc} 0 & 0 & -m-l \end{array}\right)}^{T}$. $r_{i2}^{gi}={\left(\begin{array}{ccc} 0 & 0 & -m \end{array}\right)}^{T}$. $r_{i3}^{gi}={\left(\begin{array}{ccc} 0 & 0 & 0 \end{array}\right)}^{T}$.

$K_{f1}$ and $K_{f3}$ are the stiffness matrices of the Part 1 and part 2, which have been obtained through the second chapter; $K_{f2}$ is the stiffness matrix of the part 2.

So, the stiffness matrix of the measurement branch is shown as follows,
(37)$$K_{i}^{gi}=K_{i1}^{gi}+K_{i2}^{gi}+K_{i3}^{gi}$$

### 4.3. Stiffness Matrix Modeling of Six-Dimensional Force Sensor with a Circular Flexible Spherical Joint

The end position coordinate systems of measured branches are shown as Figure 4. In Figure 4, $O_{g1}-X_{g1}Y_{g1}Z_{g1}$ is the reference coordinate system located at the geometric center of the upper platform surface. $O_{gi}-X_{gi}Y_{gi}Z_{gi}$(i=1,2,3,4,5,6) is the local coordinate system of the end of the flexible spherical joint for installing the force measuring branch. $O_{g2}-X_{g2}Y_{g2}Z_{g2}$ and $O_{g3}-X_{g3}Y_{g3}Z_{g3}$ are $O_{g1}-X_{g1}Y_{g1}Z_{g1}$ rotated 120° and 240° counterclockwise around the $Z_{p}$-axis, respectively. $O_{g5}-X_{g5}Y_{g5}Z_{g5}$ and $O_{g6}-X_{g6}Y_{g6}Z_{g6}$ are $O_{g4}-X_{g4}Y_{g4}Z_{g4}$ rotated 120° and 240° counterclockwise around the $Z_{p}$-axis, respectively. In order to ensure the clarity of the image, $O_{gi}-X_{gi}Y_{gi}Z_{gi}\left(i=2,3,5,6\right)$ is not marked in Figure 4. The axes of coordinate system $O_{p}-X_{p}Y_{p}Z_{p}$ and coordinate system $O_{g1}-X_{g1}Y_{g1}Z_{g1}$ are correspondingly parallel. $Z_{p}$ is perpendicular to the upper platform and vertically upwards. $O_{g1}$ is on the x-axis of the reference coordinate system $O_{p}-X_{p}Y_{p}Z_{p}$. ${O^{\prime }}_{p}$ is the projection of $O_{p}$ on the plane formed by three horizontal force branches. $r_{4}$ is the distance from ${O^{\prime }}_{p}$ to $O_{gi}-X_{gi}Y_{gi}Z_{gi}\left(i=4,5,6\right)$. The angle between the planes $X_{P}O_{P}Z_{P}$ and ${O^{\prime }}_{p}O_{g4}$ is γ. The direction of $Z_{gi\left(i=4,5,6\right)}$ is horizontal outward along the direction of the horizontal force measurement branch, the direction of $Y_{gi\left(i=4,5,6\right)}$ is the vertical force measuring platform straight up, and the direction of $X_{gi\left(i=4,5,6\right)}$ is determined according to the right-hand rule. The distance between the upper platform and the lower platform is l, and the thickness of the upper and lower platforms is m.

The stiffness of the measurement branch can be obtained by Equation (37).
(38)$$K_{gi}=K_{i}^{gi}=K_{i1}^{gi}+K_{i2}^{gi}+K_{i3}^{gi}$$

According to the literature, the overall stiffness matrix of the six-dimensional force sensor can be obtained as follows,
(39)$$K=\sum_{i=1}^{6}T_{gi}^{p}K_{gi}{\left(T_{gi}^{p}\right)}^{-1}$$
where $T_{gi}^{p}\left(i=1,2,\ldots ,6\right)$ represents the attitude transformation matrix of the local coordinate system at the end of the force measuring branch relative to the reference coordinate system of the upper platform. $r_{gi}^{p}\left(i=1,2,\ldots ,6\right)$ represents the coordinates of the origin of the local coordinate system relative to the reference coordinate system. $R_{gi}^{p}\left(i=1,2,\ldots ,6\right)$ represents the rotation matrix of the local coordinate system with respect to the reference coordinate system. Here,
$$\begin{array}{l} r_{g1}^{p}={\left(\begin{array}{ccc} r_{1} & 0 & 0 \end{array}\right)}^{T},R_{g1}^{p}=\left[\begin{array}{ccc} 1 & 0 & 0\\ 0 & 1 & 0\\ 0 & 0 & 1 \end{array}\right]\\ r_{g2}^{p}={\left(\begin{array}{ccc} \frac{-r_{1}}{2} & \frac{\sqrt{3}r_{1}}{2} & 0 \end{array}\right)}^{T},R_{g2}^{p}=\left[\begin{array}{ccc} -1/2 & \sqrt{3}/2 & 0\\ -\sqrt{3}/2 & -1/2 & 0\\ 1 & 1 & 1 \end{array}\right]\\ r_{g3}^{p}={\left(\begin{array}{ccc} -\frac{r_{1}}{2} & -\frac{\sqrt{3}r_{1}}{2} & 0 \end{array}\right)}^{T},R_{g3}^{p}=\left[\begin{array}{ccc} -1/2 & -\sqrt{3}/2 & 0\\ \sqrt{3}/2 & 1/2 & 0\\ 0 & 0 & 1 \end{array}\right]\\ r_{g4}^{p}={\left(\begin{array}{ccc} r_{4}\cos\left(\gamma \right) & r_{4}\sin\left(\gamma \right) & -l_{1} \end{array}\right)}^{T},R_{g4}^{p}=\left[\begin{array}{ccc} 1/2 & 0 & \sqrt{3}/2\\ -\sqrt{3}/2 & 0 & 1/2\\ 0 & -1 & 0 \end{array}\right]\\ r_{g5}^{p}={\left(\begin{array}{ccc} r_{4}\cos\left(\gamma +2\pi /3\right) & r_{4}\sin\left(\gamma +2\pi /3\right) & -l_{1} \end{array}\right)}^{T},R_{g5}^{p}=\left[\begin{array}{ccc} -1 & 0 & 0\\ 0 & 0 & -1\\ 0 & -1 & 0 \end{array}\right]\\ r_{g6}^{p}={\left(\begin{array}{ccc} r_{4}\cos\left(\gamma +4\pi /3\right) & r_{4}\sin\left(\gamma +4\pi /3\right) & -l_{1} \end{array}\right)}^{T},R_{g6}^{p}=\left[\begin{array}{ccc} 1/2 & 0 & -\sqrt{3}/2\\ \sqrt{3}/2 & 0 & 1/2\\ 0 & -1 & 0 \end{array}\right] \end{array}$$

## 5. Force Mapping Matrix Analysis of the Six-Axis Force Sensor with Circular Flexible Spherical Joints

According to literature [7,25], The relationship between the external force/torque on the upper platform of the six-dimensional force sensor and the overall deformation can be obtained as follows,
(40)F=KX
where $F={\left[\begin{array}{cccccc} F_{x} & F_{y} & F_{z} & M_{x} & M_{y} & M_{z} \end{array}\right]}^{T}$ represents the external force/torque acting on the platform of the six-dimensional force sensor. $X={\left[\begin{array}{cccccc} \Delta_{x} & \Delta_{y} & \Delta_{z} & \theta_{x} & \theta_{y} & \theta_{z} \end{array}\right]}^{T}$ represents the overall deformation of the platform on the six-dimensional force sensor.

Since the stiffness matrix *K* is a non-singular matrix, there is an inverse matrix of the stiffness matrix, that is, the flexibility matrix
(41)$$C=K^{-1}$$

So Equation (40) can be transformed.
(42)X=CF

In addition, the deformation relationship between the force branch deformation and the upper measurement platform can be obtained as follows.
(43)$$X_{i}={\left(T_{gi}^{p}\right)}^{-1}X={\left(T_{gi}^{p}\right)}^{-1}K^{-1}F\left(i=1,2,3,4,5,6\right)$$
where $X_{i}$ represents the end deformation of the measurement branch i. According to Equation (33), the relationship between $X_{i}$ and the force $F_{i}$ applied on the end of the force-sensitive element is shown as follows,
(44)$$F_{i}=K_{gi}X_{i}$$

According to the principle of virtual work, the relationship between the external force $F_{ij}\left(i,j=1,2,\cdots ,6\right)$ on the *j*th part of measuring branch *i* and the external force $F_{i(i=1,2,3,4,5,6)}$ on measuring branch *i* is as follows,
(45)$$F_{ij}={\left(T_{j}^{gi}\right)}^{-1}F_{i}\left(i=1,2,3,4,5,6;j=1,2,3\right)$$

According to the Equations (40)–(45), the relationship between $F_{ij\left(i=1,2,3,4,5,6;j=1,2,3\right)}$ and F can also be obtained as follows,
(46)$$F_{ij\left(i=1,2,3,4,5,6;j=1,2,3\right)}={\left(T_{j}^{gi}\right)}^{-1}K_{gi}{\left(T_{gi}^{p}\right)}^{-1}K^{-1}F$$

When i=1,2,3,4,5,6 and j=2, It can be obtain the relationship between the end external force of part 2 and the generalized external force applied on the upper platform.

Because the six-dimensional force sensor use circular flexible spherical joints instead of the ideal ball hinge. From the structural characteristics of the circular flexible spherical joint, it can be seen that the deformation of the measurement branch is mainly axial tension and compression. Extracting the part 2 of the measuring branch and forming the force mapping matrix by Equation (46), the relationship between the axial force applied to the measuring branch and the external force applied to the platform on the six-dimensional force sensor can be obtained, and the specific relationship can also be obtained as follows,
(47)$$\left[\begin{array}{c} F_{12}^{Z}\\ F_{22}^{Z}\\ F_{32}^{Z}\\ F_{42}^{Z}\\ F_{52}^{Z}\\ F_{62}^{Z} \end{array}\right]=\left[\begin{array}{c} {\left({\left(T_{2}^{g1}\right)}^{-1}K_{g1}{\left(T_{g1}^{p}\right)}^{-1}K^{-1}\right)}_{\left(3\right)}\\ {\left({\left(T_{2}^{g2}\right)}^{-1}K_{g2}{\left(T_{g2}^{p}\right)}^{-1}K^{-1}\right)}_{\left(3\right)}\\ {\left({\left(T_{2}^{g3}\right)}^{-1}K_{g3}{\left(T_{g3}^{p}\right)}^{-1}K^{-1}\right)}_{\left(3\right)}\\ {\left({\left(T_{2}^{g4}\right)}^{-1}K_{g4}{\left(T_{g4}^{p}\right)}^{-1}K^{-1}\right)}_{\left(3\right)}\\ {\left({\left(T_{2}^{g5}\right)}^{-1}K_{g5}{\left(T_{g5}^{p}\right)}^{-1}K^{-1}\right)}_{\left(3\right)}\\ {\left({\left(T_{2}^{g6}\right)}^{-1}K_{g6}{\left(T_{g6}^{p}\right)}^{-1}K^{-1}\right)}_{\left(3\right)} \end{array}\right]\left[\begin{array}{c} F_{x}\\ F_{y}\\ F_{z}\\ M_{x}\\ M_{y}\\ M_{z} \end{array}\right]$$
where i=1,2,3,4,5,6, $F_{i2}^{Z}$ represents the axial force of part 2 of the measurement branch. ${\left({\left(T_{2}^{gi}\right)}^{-1}K_{gi}{\left(T_{gi}^{p}\right)}^{-1}K^{-1}\right)}_{\left(3\right)}$ denotes the third row of the matrix ${\left({\left(T_{2}^{gi}\right)}^{-1}K_{gi}{\left(T_{gi}^{p}\right)}^{-1}K^{-1}\right)}_{6*6}$. Equation (47) can be simplified as follows,
(48)$$F=GF_{i}$$
where $F_{i}$ denotes the axial force of part 2 of the measurement branch. G denotes the force mapping matrix between *f_i_* and F.

The force mapping matrix *G* of the sensor is represented as follows,
(49)$$G={\left[\begin{array}{c} {\left({\left(T_{2}^{g1}\right)}^{-1}K_{g1}{\left(T_{g1}^{p}\right)}^{-1}K^{-1}\right)}_{\left(3\right)}\\ {\left({\left(T_{2}^{g2}\right)}^{-1}K_{g2}{\left(T_{g2}^{p}\right)}^{-1}K^{-1}\right)}_{\left(3\right)}\\ {\left({\left(T_{2}^{g3}\right)}^{-1}K_{g3}{\left(T_{g3}^{p}\right)}^{-1}K^{-1}\right)}_{\left(3\right)}\\ {\left({\left(T_{2}^{g4}\right)}^{-1}K_{g4}{\left(T_{g4}^{p}\right)}^{-1}K^{-1}\right)}_{\left(3\right)}\\ {\left({\left(T_{2}^{g5}\right)}^{-1}K_{g5}{\left(T_{g5}^{p}\right)}^{-1}K^{-1}\right)}_{\left(3\right)}\\ {\left({\left(T_{2}^{g6}\right)}^{-1}K_{g6}{\left(T_{g6}^{p}\right)}^{-1}K^{-1}\right)}_{\left(3\right)} \end{array}\right]}^{-1}$$

## 6. The Influence of the Circular Flexible Spherical Joints on the Accuracy of Six-Dimensional Force Sensor

The circular flexible spherical joint has many structural parameters, except for structural parameter l,t,R, the rest of the parameters are related to the design size of the sensor and the use of the requirements, so this paper only considers the effect of structural parameter l,t,R on the accuracy of the sensor. The force mapping matrices of the six-dimensional force sensors applying the ideal ball hinge and the circular flexible spherical joint are calculated separately, and the influence of the structural parameters on the accuracy of the sensors is judged by comparing the degree of approximation of the two. When the force mapping matrix of the six-dimensional force sensor applying the circular flexible spherical joint is closer to that of the ideal ball hinge, it indicates that the coupling error of the sensor is smaller and the accuracy is higher; otherwise, the coupling error of the sensor is larger and the accuracy is lower. In order to facilitate the calculation of the force mapping matrix of the sensor, the values of the parameters are set to the specified parameter values except for the structural parameter l,t,R, as shown in Table 3.

Because the force mapping matrix for six-dimensional force sensor with circular flexible spherical joints has been established in the previous sections, the force mapping matrix for applying an ideal ball-hinged six-dimensional force sensor also needs to be established. As shown in Figure 3, the coordinate parameters $a_{i}^{O}$ and $A_{i}^{O}$(i=1,2,3,4,5,6) of the sensor are as follows.
(50)$$A=\left[\begin{array}{cccccc} 0.072 & -0.036 & -0.036 & 0.025\sqrt{3} & -0.025\sqrt{3} & 0\\ 0 & 0.036\sqrt{3} & -0.036\sqrt{3} & 0.025 & 0.025 & -0.05\\ -0.05 & -0.05 & -0.05 & -0.025 & -0.025 & -0.025 \end{array}\right]$$
(51)$$a=\left[\begin{array}{cccccc} 0.072 & -0.036 & -0.036 & 0 & -0.025\sqrt{3} & 0.025\sqrt{3}\\ 0 & 0.036\sqrt{3} & -0.036\sqrt{3} & 0.05 & -0.025 & -0.025\\ 0 & 0 & 0 & -0.025 & -0.025 & -0.025 \end{array}\right]$$

According to the Equation (22), the force mapping matrix $G_{r}$ can be derived as follows,
(52)$$G_{r}=\left[\begin{array}{cccccc} 0 & 0 & 0 & -\frac{\sqrt{3}}{2} & 0 & \frac{\sqrt{3}}{2}\\ 0 & 0 & 0 & \frac{1}{2} & -1 & \frac{1}{2}\\ 1 & 1 & 1 & 0 & 0 & 0\\ 0 & 0.036\sqrt{3} & -0.036\sqrt{3} & 0.0125 & 0 & 0.0125\\ 0.072 & 0.036 & 0.036 & 0.0125\sqrt{3} & -0.05 & -0.0125\sqrt{3}\\ 0 & 0 & 0 & 0.025\sqrt{3} & 0.025\sqrt{3} & 0.025\sqrt{3} \end{array}\right]$$

The force mapping matrix of the sensor with the novel circular flexible spherical joints can be obtained according to the parameters in Table 3. In order to easily see the effect of *l* on accuracy, the physical quantity θ is deliberately introduced. θ is one-half of the central angle corresponding to the arc with radius *R*. When *R* is constant. According to the geometric relationship of the circular flexible spherical joint, the relationship of the parameters between *l* and θ can be derived as follows,
(53)l=2Rsinθ

When *R* is constant and θ<π/2, l increases as θ increases.

According to the iterative assignment method, the iterative parameters are selected as follows: the value range of the parameter t is from 0.002 to 0.0065 m and the parameter interval is 0.0005 m. The value range of the parameter *R* is from 0.0003 to 0.0015 m and the parameter interval is 0.0005 m. The value range of the parameter θ is from 0.1 to 0.5*π* and the parameter interval is 0.08*π*.

To analyze the accuracy of the parameters of the circular flexible spherical joint structure on the sensor, the Euclidean distance physical quantity [26,27,28,29] is hereby introduced. Euclidean distance is a more widely defined distance that refers to the true distance between two points in M-dimensional space, or the natural length of a vector. A Euclidean distance in two- or three-dimensional space is the distance between two points. For two matrices with the same dimension, the Euclidean distance between the matrices indicates the closeness of the matrix, the larger the Euclidean distance, the lower the closeness, and the smaller the Euclidean distance, the higher the closeness. The Euclidean distance formula for two matrices of the same dimension with a specific number of rows is as follows:$$\begin{array}{c} A=\left(\begin{array}{c} a_{1}\\ a_{2} \end{array}\right)=\left[\begin{array}{ccc} a_{11} & a_{12} & a_{13}\\ a_{21} & a_{22} & a_{23} \end{array}\right]\\ B=\left(\begin{array}{c} b_{1}\\ b_{2} \end{array}\right)=\left[\begin{array}{ccc} b_{11} & b_{12} & b_{13}\\ b_{21} & b_{22} & b_{23} \end{array}\right] \end{array}$$

Matrix A and B will form a matrix C, the dimension of matrix C is 2×3, the mathematical relationship can be obtained as follows,
$$\begin{array}{l} C=\left(\begin{array}{c} c_{1}\\ c_{2} \end{array}\right)=\left[\begin{array}{ccc} c_{11} & c_{12} & c_{13}\\ c_{21} & c_{22} & c_{23} \end{array}\right]\\ c_{11}=\sqrt{{\left(a_{11}-b_{11}\right)}^{2}+{\left(a_{12}-b_{12}\right)}^{2}+{\left(a_{13}-b_{13}\right)}^{2}}={\left\|a_{1}-b_{1}\right\|}_{2}\\ c_{12}=\sqrt{{\left(a_{11}-b_{21}\right)}^{2}+{\left(a_{12}-b_{22}\right)}^{2}+{\left(a_{13}-b_{23}\right)}^{2}}={\left\|a_{1}-b_{2}\right\|}_{2}\\ c_{13}=\sqrt{{\left(a_{11}-b_{31}\right)}^{2}+{\left(a_{12}-b_{32}\right)}^{2}+{\left(a_{13}-b_{33}\right)}^{2}}={\left\|a_{1}-b_{3}\right\|}_{2}\\ c_{21}=\sqrt{{\left(a_{21}-b_{11}\right)}^{2}+{\left(a_{22}-b_{12}\right)}^{2}+{\left(a_{23}-b_{13}\right)}^{2}}={\left\|a_{2}-b_{1}\right\|}_{2}\\ c_{22}=\sqrt{{\left(a_{21}-b_{21}\right)}^{2}+{\left(a_{22}-b_{22}\right)}^{2}+{\left(a_{23}-b_{23}\right)}^{2}}={\left\|a_{2}-b_{2}\right\|}_{2}\\ c_{23}=\sqrt{{\left(a_{21}-b_{31}\right)}^{2}+{\left(a_{22}-b_{32}\right)}^{2}+{\left(a_{23}-b_{33}\right)}^{2}}={\left\|a_{2}-b_{3}\right\|}_{2} \end{array}$$

The rightmost equation also expresses the *L*2-norm of a vector.

Euclidean distance can be obtained as follows,
$$\rho =\sqrt{{\left(c_{11}-c_{21}\right)}^{2}+{\left(c_{12}-c_{22}\right)}^{2}+{\left(c_{31}-c_{32}\right)}^{2}}$$

According to the above expression, it can be deduced that the formula for calculating the Euclidean distance between two matrices with the same dimension can be derived as follows,

Suppose the size of the test set matrix P is M×D (there are M points in the test set, and each point is D-dimensional feature vector), and the size of the matrix C is N×D (there are N points in the test set, and each point is D dimensional feature vector), $P_{i}$ is the i line of P, $C_{j}$ is the i line of C.
$$P_{i}=\left[\begin{array}{cccc} P_{i1} & P_{i2} & \cdots & P_{iD} \end{array}\right],C_{j}=\left[\begin{array}{cccc} C_{j1} & C_{j2} & \cdots & C_{jD} \end{array}\right]$$

The Euclidean distance ρ(i,j) between $P_{i}$ and $C_{j}$ can be obtained as follows,
$$\begin{array}{l} \rho \left(i,j\right)=\sqrt{{\left(P_{i1}-C_{i1}\right)}^{2}+{\left(P_{i2}-C_{i2}\right)}^{2}+\cdots +{\left(P_{iD}-C_{iD}\right)}^{2}}\\ =\sqrt{\left(P_{i1}^{2}+P_{i2}^{2}+\cdots +P_{iD}^{2}\right)+\left(C_{j1}^{2}+C_{j2}^{2}+\cdots +C_{jD}^{2}\right)-2\times \left(P_{i1}C_{j1}+P_{i2}C_{j2}+\cdots +P_{iD}C_{jD}\right)}\\ =\sqrt{{\left\|P_{i}\right\|}^{2}+{\left\|C_{j}\right\|}^{2}-2P_{i}C_{j}^{T}} \end{array}$$

The Euclidean distance calculation formula that can be extended to the *i*-th row from the above formula can be derived as follows,
$$\rho \left(i\right)=\sqrt{\left(\begin{array}{cccc} {\left\|P_{i}\right\|}^{2} & {\left\|P_{i}\right\|}^{2} & \cdots & {\left\|P_{i}\right\|}^{2} \end{array}\right)+\left(\begin{array}{cccc} {\left\|C_{1}\right\|}^{2} & {\left\|C_{2}\right\|}^{2} & \cdots & {\left\|C_{N}\right\|}^{2} \end{array}\right)-2\times P_{i}C^{T}}$$

So the Euclidean distance between matrix P and matrix C can be derived as follows,
$$\rho =\sqrt{\left(\begin{array}{cccc} {\left\|P_{1}\right\|}^{2} & {\left\|P_{1}\right\|}^{2} & \cdots & {\left\|P_{1}\right\|}^{2}\\ {\left\|P_{2}\right\|}^{2} & {\left\|P_{2}\right\|}^{2} & \cdots & {\left\|P_{2}\right\|}^{2}\\ \vdots & \vdots & \ddots & \vdots \\ {\left\|P_{1}\right\|}^{2} & {\left\|P_{1}\right\|}^{2} & \cdots & {\left\|P_{1}\right\|}^{2} \end{array}\right)+\left(\begin{array}{cccc} {\left\|C_{1}\right\|}^{2} & {\left\|C_{2}\right\|}^{2} & \cdots & {\left\|C_{N}\right\|}^{2}\\ {\left\|C_{1}\right\|}^{2} & {\left\|C_{2}\right\|}^{2} & \cdots & {\left\|C_{N}\right\|}^{2}\\ \vdots & \vdots & \ddots & \vdots \\ {\left\|C_{1}\right\|}^{2} & {\left\|C_{2}\right\|}^{2} & \cdots & {\left\|C_{N}\right\|}^{2} \end{array}\right)-2\times PC^{T}}$$

Using the Euclidean distance to compare the similarity of the force mapping matrices $G_{r}$ and G. The smaller the Euclidean distance, the higher the similarity between the two matrices, that is, the more accurate the six-dimensional force sensor. Figure 8 shows the three-dimensional relationship between Euclidean distance and R,t,θ. Figure 9 shows the relationship between Euclidean distance and t. Figure 10 shows the relationship between Euclidean distance and R.

As shown in Figure 8, Figure 9 and Figure 10. The following conclusions can be obtained:
According to Figure 8, when *R* and *t* are constant, the larger θ is, the smaller Euclidean distance of the difference matrix is, similarly the smaller θ is, the larger Euclidean distance of the difference matrix is;According to Figure 9, when *R* and θ are constant, the larger *t* is, the larger Euclidean distance of the difference matrix is, similarly the smaller *t* is, the smaller Euclidean distance of the difference matrix is;According to Figure 10, when *t* and θ are constant, the larger *R* is, the smaller Euclidean distance of the difference matrix is, similarly the smaller *R* is, the larger Euclidean distance of the difference matrix is;The parameter *t* has the largest influence on Euclidean distance, followed by *R*, and the influence of θ is the smallest.


## 7. Instance Verification

In order to verify the above analysis results, this paper uses the finite element method to verify the above results. First, two 3D models of six-dimensional force sensors with different Euclidean distances are established, and their structural parameters were identical except for the structural parameters analyzed above. Second, the calibration matrix of the six-dimensional force sensor is established through finite element simulation software, the error matrix of the six-dimensional force sensor is established, and the linearity error and the maximum coupling error are selected to comment on the above analysis results. Finally, a six-dimensional force sensor prototype was developed, its calibration matrix was deduced through experiments, and its error is analyzed, which verifies that the developed six-dimensional force sensor had high precision.

### 7.1. Verification through the Finite Element Method

In order to verify the correctness of the above analysis results, a three-dimensional model of sensor 1 with *l* = 6 mm, *R* = 4 mm, and *t* = 2 mm was established, and a three-dimensional model of sensor 2 with *l* = 4 mm, *R* = 4 mm, and *t* = 2 mm was established, abbreviated as sensor 1 and sensor 2. Figure 11 shows the stress cloud diagram of sensor 2 when the torque is 5 N∙m.

In order to verify the accuracy of the above analysis results, the calibration matrix of the sensor is obtained from the finite element simulation results using the least squares fitting principle, and the specific process is expressed in mathematical formulas as follows,
(54)$$F_{s}=GU$$

Here,
$$\begin{array}{l} F_{s}=\left[\begin{array}{cccccccccccc} F_{x} & \cdots & 0 & 0 & 0 & 0 & 0 & 0 & 0 & 0 & 0 & 0\\ 0 & 0 & F_{y} & \ldots & 0 & 0 & 0 & 0 & 0 & 0 & 0 & 0\\ 0 & 0 & 0 & 0 & F_{z} & \ldots & 0 & 0 & 0 & 0 & 0 & 0\\ 0 & 0 & 0 & 0 & 0 & 0 & M_{x} & \ldots & 0 & 0 & 0 & 0\\ 0 & 0 & 0 & 0 & 0 & 0 & 0 & 0 & M_{y} & \ldots & 0 & 0\\ 0 & 0 & 0 & 0 & 0 & 0 & 0 & 0 & 0 & 0 & M_{z} & \ldots \end{array}\right];\\ U=\left[\begin{array}{cccccccccccc} U_{Fx1} & \cdots & U_{Fy1} & \cdots & U_{Fz1} & \cdots & U_{Mx1} & \cdots & U_{My1} & \cdots & U_{Mz1} & \cdots \\ U_{Fx2} & \cdots & U_{Fy2} & \cdots & U_{Fz2} & \cdots & U_{Mx2} & \cdots & U_{My2} & \cdots & U_{Mz2} & \cdots \\ U_{Fx3} & \cdots & U_{Fy3} & \cdots & U_{Fz3} & \cdots & U_{Mx3} & \cdots & U_{My3} & \cdots & U_{Mz3} & \cdots \\ U_{Fx4} & \cdots & U_{Fy4} & \cdots & U_{Fz4} & \cdots & U_{Mx4} & \cdots & U_{My4} & \cdots & U_{Mz4} & \cdots \\ U_{Fx5} & \cdots & U_{Fy5} & \cdots & U_{Fz5} & \cdots & U_{Mx5} & \cdots & U_{My5} & \cdots & U_{Mz5} & \cdots \\ U_{Fx6} & \cdots & U_{Fy6} & \cdots & U_{Fz6} & \cdots & U_{Mx6} & \cdots & U_{My6} & \cdots & U_{Mz6} & \cdots \end{array}\right]; \end{array}$$
where $F_{s}$ is the loading on moving platform, *G* is the calibration matrix, U is the output voltage value.

For Equation (54), The calibration matrix *G* can be obtained by the following equation, which takes the form shown below.
(55)$$G=F_{s}U^{-}$$
where $U^{-}$ is the pseudoinverse matrix of U, $U^{-}=U^{T}{\left(UU^{T}\right)}^{-1}$. The matrix of *G* can be obtained as follows,
(56)$$G=F_{s}U^{T}{\left(UU^{T}\right)}^{-1}$$

$G_{6\times n}$ is calibration matrix, which is also a decoupling matrix. In order to obtain a more accurate calibration matrix, the number of calibrations must be much greater than the dimension of the calibration matrix.

After obtaining the calibration matrix of the sensor, the error matrix E can be obtained through its measured value. The error matrix is performance index for judging the accuracy of the sensor and the coupling error. The calculation formula of the error matrix can be obtained as follows,
(57)$$E=\left|F_{s}-F_{c}\right|F_{m}^{-1}$$

Here, $\left|F_{s}-F_{c}\right|$ is the difference between $F_{s}$ and $F_{c}$. $F_{m}=diag\left(\begin{array}{cccccc} F_{x\max} & F_{y\max} & F_{z\max} & M_{x\max} & M_{y\max} & M_{z\max} \end{array}\right)$ is a diagonal matrix of the corresponding dimensional force/torque full-scale values. $F_{s}$ is the actual loading value of the six-dimensional force sensor. $F_{c}=GU$ is the actual measured value of the sensor.

The calibration matrices of sensor 1 and sensor 2 calculated by the above formula are $G_{1}$ and $G_{2}$ can be obtained as follows,
(58)$$G_{1}=\left[\begin{array}{cccccc} 0.5435 & 0.0147 & -2.0937 & 0.0051 & 0.0116 & 0.0393\\ 0.0462 & -0.2535 & -0.3227 & -0.0003 & 0.0245 & \;0.0167\\ -0.2043 & \;0.9910 & 1.0028 & 0.0016 & -0.0170 & 0.0026\\ 0.0045 & -0.0417 & -0.0322 & -0.0000 & 0.0008 & 0.0022\\ 0.0093 & 0.0911 & 0.3108 & -0.0003 & -0.0036 & -0.0056\\ 0.0002 & 0.2147 & 0.0320 & -0.0001 & -0.0043 & 0.0004 \end{array}\right]\times 10^{8}$$
(59)$$G_{2}=\left[\begin{array}{cccccc} 0.6992 & -0.7059 & -0.2064 & -0.0038 & 0.0150 & 0.0008\\ 0.8790 & -2.0804 & 0.0126 & -0.0150 & 0.0551 & 0.0023\\ -1.7013 & 4.2644 & -0.0825 & 0.0293 & -0.0719 & 0.0355\\ 0.1036 & -0.2564 & 0.0006 & -0.0017 & 0.0047 & 0.0003\\ 0.0253 & 0.0842 & 0.0164 & 0.0003 & -0.0017 & 0\\ 0.0709 & 0.0147 & -0.0026 & -0.0013 & -0.0002 & 0.0001 \end{array}\right]\times 10^{8}$$

After analysis, the error matrices $E_{L1}$ and $E_{L2}$ of sensor 1 and sensor 2 can be obtained as follows,
(60)$$E_{L1}=\left[\begin{array}{cccccc} 0.0048 & 0.0091 & 0.0074 & 0.0050 & 0.0051 & 0.0348\\ 0.0031 & 0.0118 & 0.0247 & 0.0171 & 0.0171 & 0.0059\\ 0.0135 & 0.016 & 0.0478 & 0.0329 & 0.0329 & 0.0429\\ 0.0046 & 0.0068 & 0.0033 & 0.0078 & 0.0155 & 0.0072\\ 0.0035 & 0.0076 & 0.0015 & 0.0095 & 0.0091 & 0.0060\\ 0.0074 & 0.0045 & 0.0018 & 0.0082 & 0.0094 & 0.0011 \end{array}\right]$$
(61)$$E_{L2}=\left[\begin{array}{cccccc} 0.0035 & 0.0035 & 0.0066 & 0.0027 & 0.0035 & 0.0126\\ 0.0024 & 0.0063 & 0.0139 & 0.0084 & 0.0093 & 0.0004\\ 0.0068 & 0.0068 & 0.0183 & 0.0112 & 0.0110 & 0.0017\\ 0.0027 & 0.0029 & 0.0013 & 0.0008 & 0.0107 & 0.0050\\ 0.0046 & 0.0061 & 0.0061 & 0.0082 & 0.0082 & 0.0029\\ 0.0021 & 0.0013 & 0.0009 & 0.0069 & 0.0069 & 0.0002 \end{array}\right]$$

The analysis results show that the maximum linearity error of sensor 1 is 4.78%, and the maximum coupling error is 4.29%; the maximum linearity error of sensor 2 is 1.83%, and the maximum coupling error is 1.26%. The linearity error and the maximum coupling error are reduced by 2.95% and 3.03%, respectively. It can be seen that the smaller the Euclidean distance, the higher the accuracy of the sensor.

### 7.2. Verification through the Experimental Method

To verify the measurement accuracy of the sensor, we selected suitable data and developed a prototype of a novel circular flexible spherical joint six-dimensional force sensor. The prototype is shown in Figure 12. Because the tension and pressure sensor has high measurement accuracy, the tension and pressure sensor was selected as the measurement branch, and the tension and pressure sensor of the model SBT641A is selected, its specific size was φ10×22 mm and the range was ±200 N.

In the practical application of the six-dimensional force sensor, the six-dimensional force sensor is usually installed at the wrist of the robot. At this time, the six-dimensional force sensor was in a six-degree-of-freedom motion state. To make the calibration process more in line with the actual application, a six-degree-of-freedom ABB industrial robot was selected for the calibration experiments. The experimental setup is shown in Figure 13. a six-dimensional force sensor needs to be installed on the wrist of the robot, and the sensor is connected to the transmitter, which provides it with an appropriate working voltage and output voltage change signal. Since the operating voltage of the sensor and transmitter must not exceed 24 V, the 220 V supply is converted to a 24 V supply through the adapter power. The data acquisition device collects the voltage change signal from the transmitter output and passes it to the PC port, where the computer completes data collection. simultaneously, the robot needs to be connected to a 220 V power supply, and the robot’s posture is adjusted through the teaching pendant to complete the calibration experiment for the six directions of the six-dimensional force sensor. During the calibration process, ten loading points were set in the direction to be calibrated, and loading and unloading experiments were carried out sequentially. Take the average of the measured experimental data at each loading point and calculate the standard deviation. The smaller the standard deviation, the better the actual measurement conditions. The interval between the force/moment loading values was 10 N/250 N∙mm, that is, the direction to be calibrated was loaded and unloaded 10 times, each time adding or subtracting a 1 kg weight. The calibration procedure was repeated to obtain the output voltage corresponding to the loaded value. The z-direction loading experiment is shown in Figure 14. To judge the accuracy of the six-dimensional force sensor, it is necessary to obtain the calibration and error matrices. In this paper, the least squares method is used to obtain the calibration matrix, and the principle is described in Equations (54)–(56).

According to the least squares fitting principle, the resulting calibration matrix *G* can be obtained as follows,
(62)$$G=\left[\begin{array}{cccccc} 0.3027 & -0.0117 & 0.2682 & -0.1849 & -0.1881 & 0.2915\\ -0.1824 & -0.3036 & -0.0517 & -0.0382 & -0.1014 & -0.1283\\ -0.0082 & 0.0505 & -0.4996 & -0.4869 & -0.4912 & -0.5582\\ -0.0451 & -0.0774 & -0.0272 & -0.0250 & -0.0121 & -0.0171\\ -0.0761 & 0.0026 & -0.0402 & 0.0175 & 0.0180 & -0.0437\\ -0.0017 & -0.0004 & -0.0082 & 0.0071 & -0.0062 & 0.0059 \end{array}\right]$$

After obtaining the calibration matrix of the sensor, the error matrix *E* can be obtained according to the above Equation (57). By loading the experiment, the load value $F_{s}$ and the sensor measured value $F_{c}$ can be obtained as follows,
(63)$$F_{s}=\left[\begin{array}{cccccc} 10\;N & 0 & 0 & 0 & 0 & 0\\ 0 & 10\;N & 0 & 0 & 0 & 0\\ 0 & 0 & 10\;N & 0 & 0 & 0\\ 0 & 0 & 0 & 2\;Nm & 0 & 0\\ 0 & 0 & 0 & 0 & 2\;Nm & 0\\ 0 & 0 & 0 & 0 & 0 & 2\;Nm \end{array}\right]$$
(64)$$F_{c}=\left[\begin{array}{cccccc} 10.0328\;N & 1.5399\;N & 0.7271\;N & 0.6886\;N & 0.4871\;N & 0.1506\;N\\ -0.1819\;N & 9.4746\;N & -1.0240\;N & -1.480\;N & -0.6335\;N & -0.2939\;N\\ -0.1588\;N & -0.8026\;N & 9.6120\;N & -2.4533\;N & -1.0845\;N & 0.6591\;N\\ -0.0408\;Nm & -0.3761\;Nm & -0.045\;Nm & 1.8623\;Nm & -0.2515\;Nm & -0.0474\;Nm\\ -0.0045\;Nm & -0.7601\;Nm & -0.2173\;Nm & -0.1948\;Nm & 1.8900\;Nm & -0.0217\;Nm\\ 0.0158\;Nm & 0.0132\;Nm & 0.0203\;Nm & 0.0432\;Nm & 0.0336\;Nm & 1.9789\;Nm \end{array}\right]$$

The error matrix *E* can be obtained as follows,
(65)$$E=\left[\begin{array}{cccccc} 0.0032 & 0.0117 & 0.092 & 0.053 & 0.0049 & 0.0014\\ 0.0019 & 0.0061 & 0.0131 & 0.0136 & 0.0062 & 0.0021\\ 0.0011 & 0.0104 & 0.0042 & 0.0232 & 0.0114 & 0.0061\\ 0.0005 & 0.0191 & 0.0005 & 0.0053 & 0.0104 & 0.0024\\ 0.0043 & 0.0501 & 0.0202 & 0.0181 & 0.0103 & 0.0022\\ 0.0009 & 0.0008 & 0.0026 & 0.0031 & 0.0029 & 0.0027 \end{array}\right]$$

According to the above error matrix, it can be found that the maximum linearity error of the prototype developed according to the structural parameters of sensor 1 is 1.03%, and the maximum coupling error is 5.01%. This result shows that the developed six-dimensional force sensor has good precision and the selected structural parameters are reasonable.

## 8. Conclusions

Firstly, we propose a circular flexible spherical joint with easy machining, high applicability and symmetrical structure, and design a six-dimensional force sensor based on it. Secondly, the stiffness matrix of a circular flexible spherical joint is derived using numerical analysis and microelement methods. Then, Through the decomposition of the sensor model, we derive the stiffness matrix of the sensor. The force mapping matrix between the force measuring element and the upper platform is established by the stiffness matrix of the sensor, the concept of Euclidean distance is introduced, and the influence of spherical joint parameters *R*, *l* and *t* on the accuracy of the sensor is analyzed by comparing the Euclidean distance of the force-mapping matrix of the circular flexible spherical joint and the ideal spherical joint. Finally, a set of optimal sensor parameters will be selected and the prototype will be developed. The following conclusions can be drawn from the finite element method and experimental method: 1. This approach is justified by comparing the Euclidean distance between the circular flexible spherical joint and the ideal spherical joint, and then judging the measurement accuracy of the sensor, this method is reasonable. 2. The calibration experiment showed that the maximum linearity error was 1.03% and the maximum coupling error was 5.01%. The results show that the developed sensor has good accuracy, and the analysis of its accuracy impact is conducive to the design and development of future sensors.

## Figures and Tables

**Figure 1 sensors-23-07247-f001:**
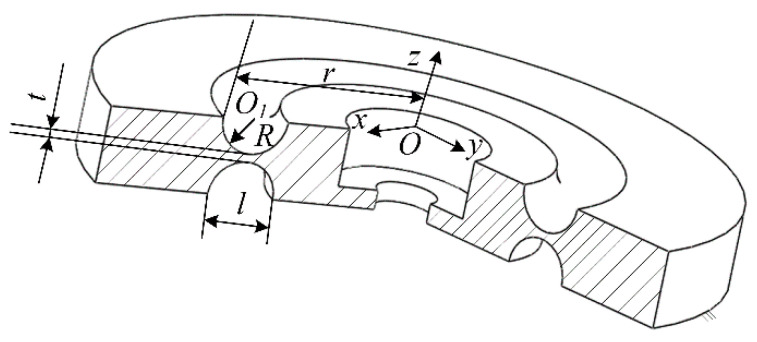
Configuration of the novel circular flexible spherical joint.

**Figure 2 sensors-23-07247-f002:**
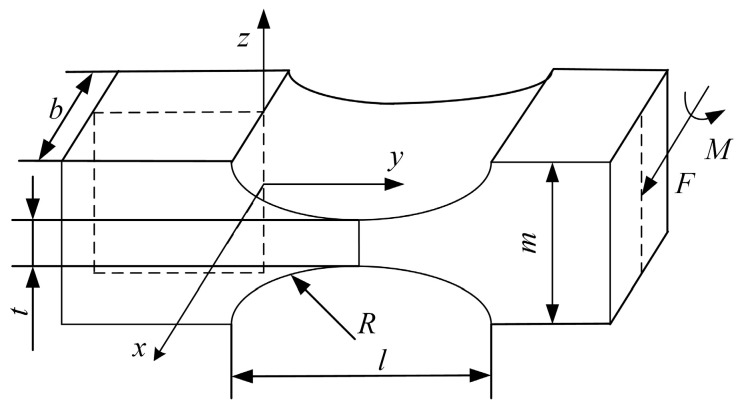
Compliant mechanism.

**Figure 3 sensors-23-07247-f003:**
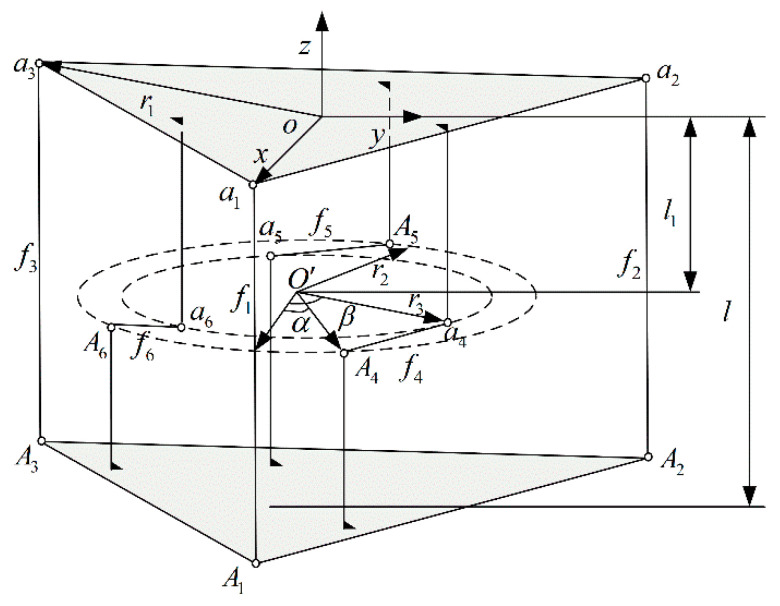
Schematic diagram of the structure of the six-dimensional force sensor.

**Figure 4 sensors-23-07247-f004:**
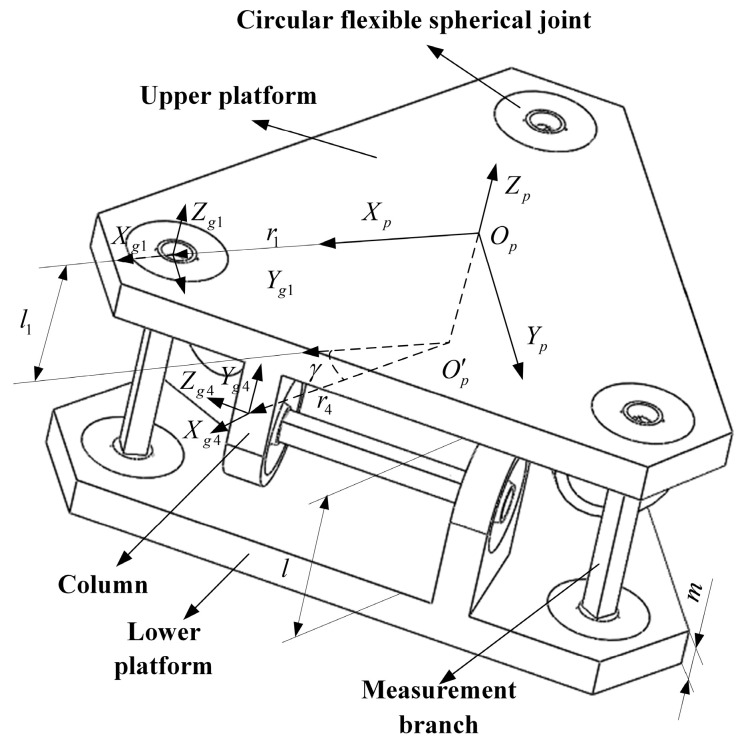
The novel circular flexible spherical joints six-dimensional force sensor.

**Figure 5 sensors-23-07247-f005:**
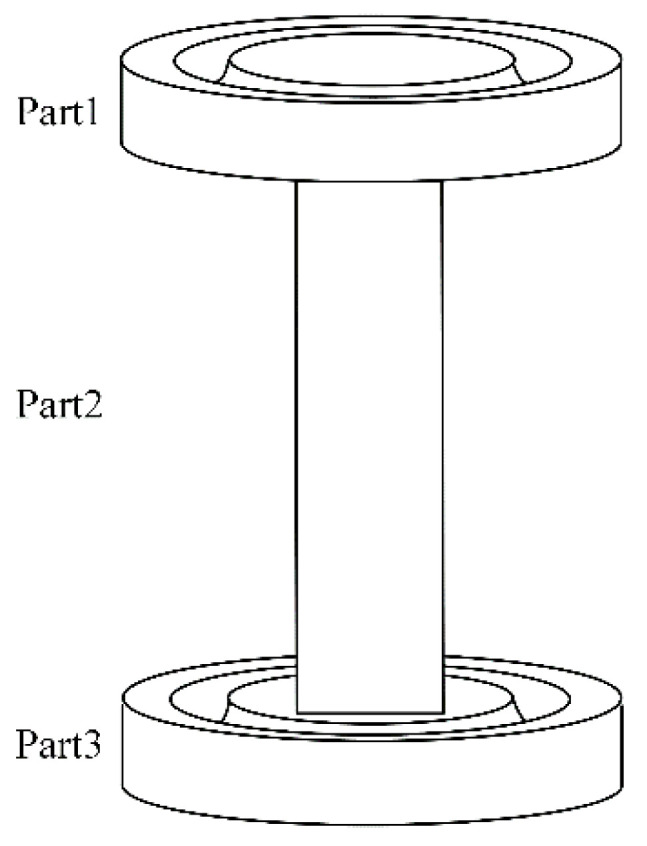
Decomposed diagram of measuring branch.

**Figure 6 sensors-23-07247-f006:**
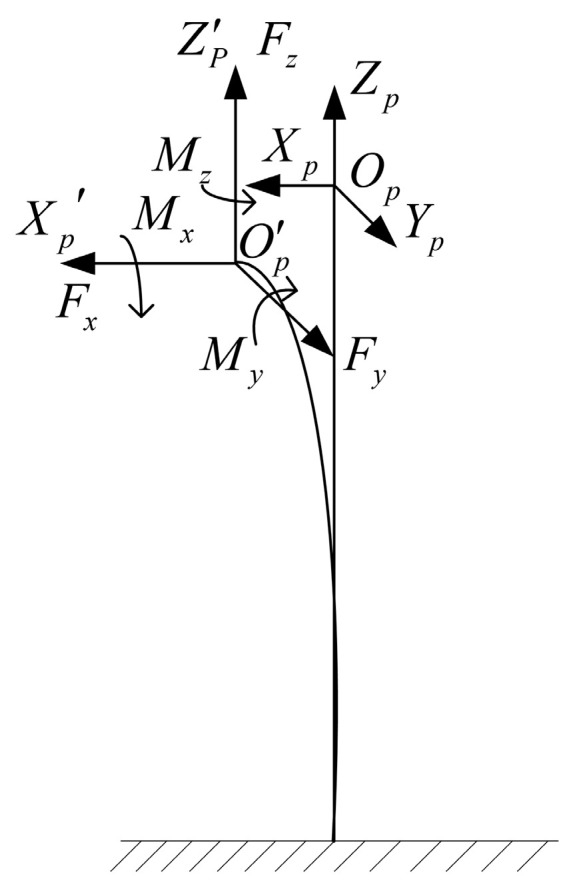
Deformation diagram of the cantilever beam under forces.

**Figure 7 sensors-23-07247-f007:**
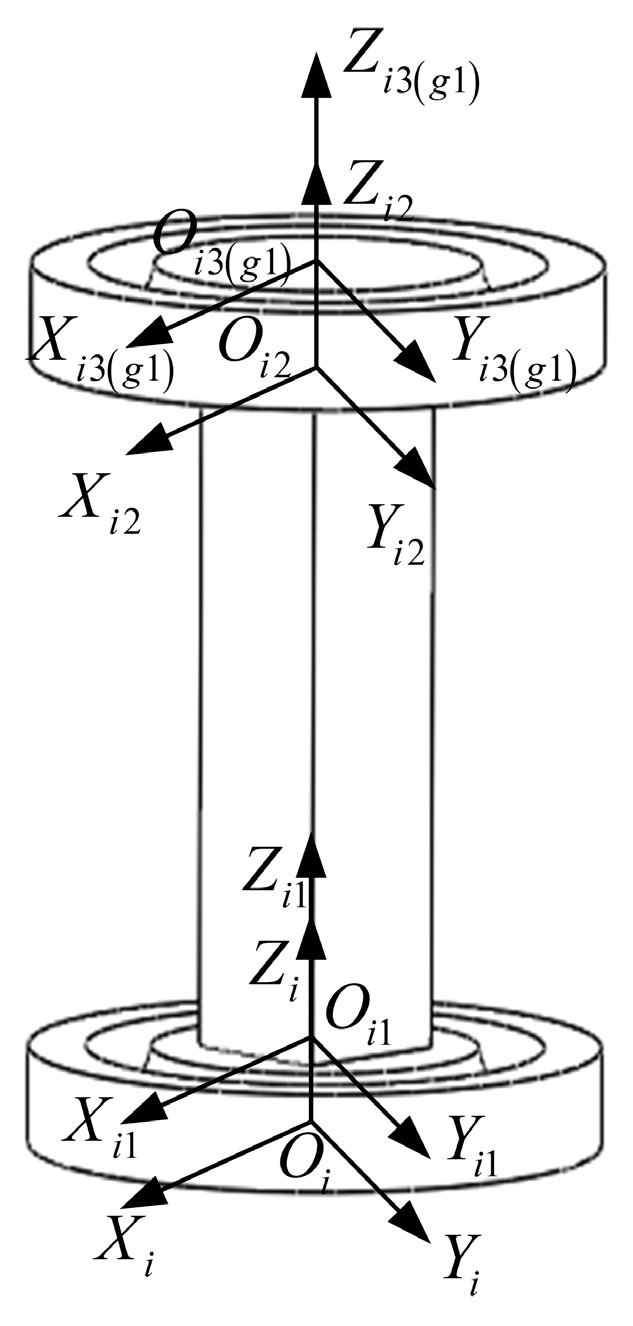
Measurement branch.

**Figure 8 sensors-23-07247-f008:**
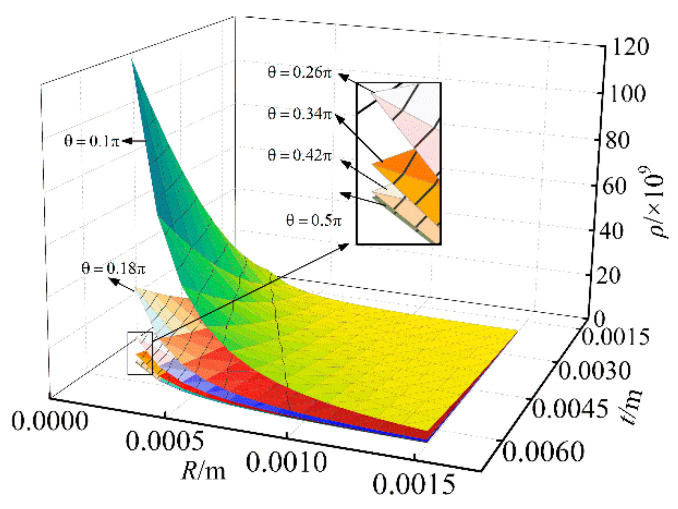
The relationship between Euclidean distance and R,t,θ.

**Figure 9 sensors-23-07247-f009:**
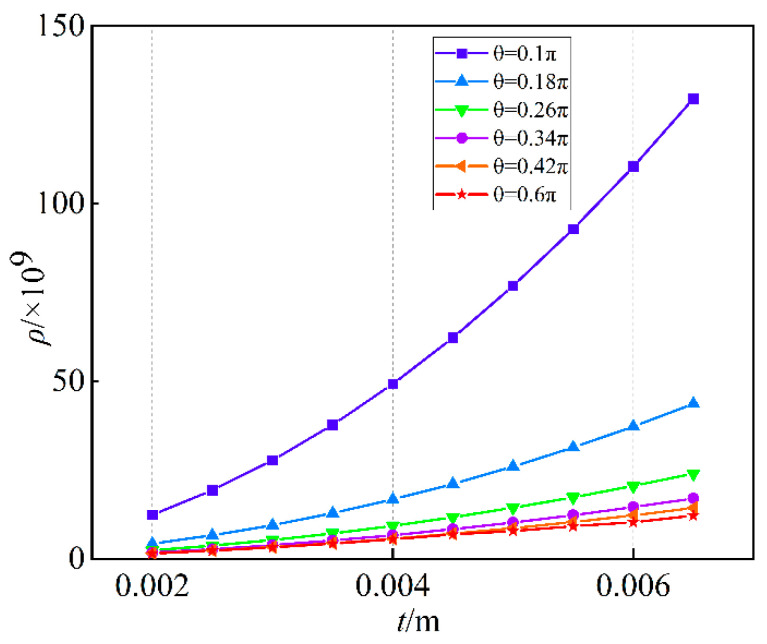
The relationship between Euclidean distance and t.

**Figure 10 sensors-23-07247-f010:**
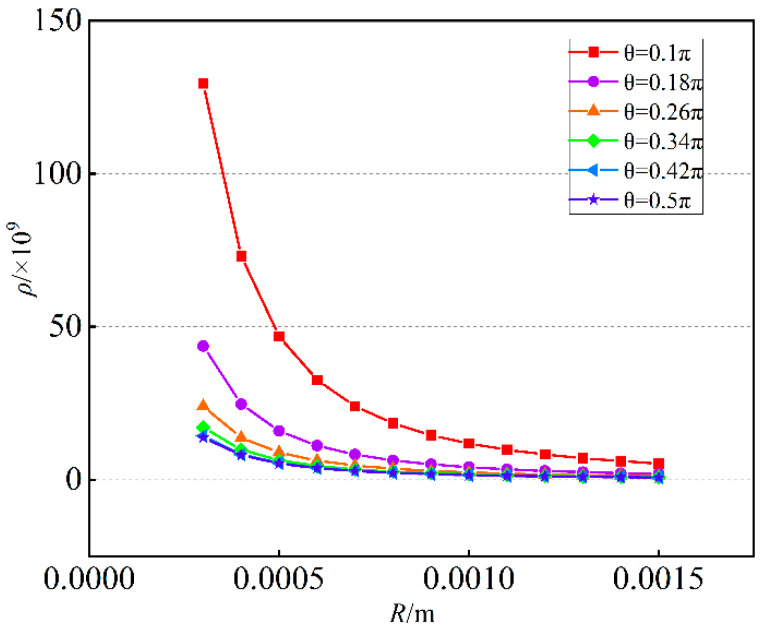
The relationship between Euclidean distance and *R*.

**Figure 11 sensors-23-07247-f011:**
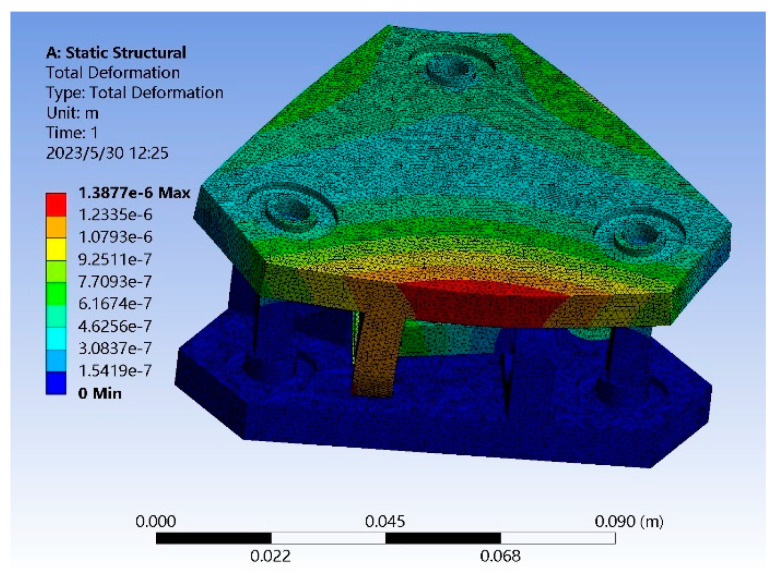
The stress cloud diagram when the torque $M_{x}=5\;N\cdot m$ applied on the sensor.

**Figure 12 sensors-23-07247-f012:**
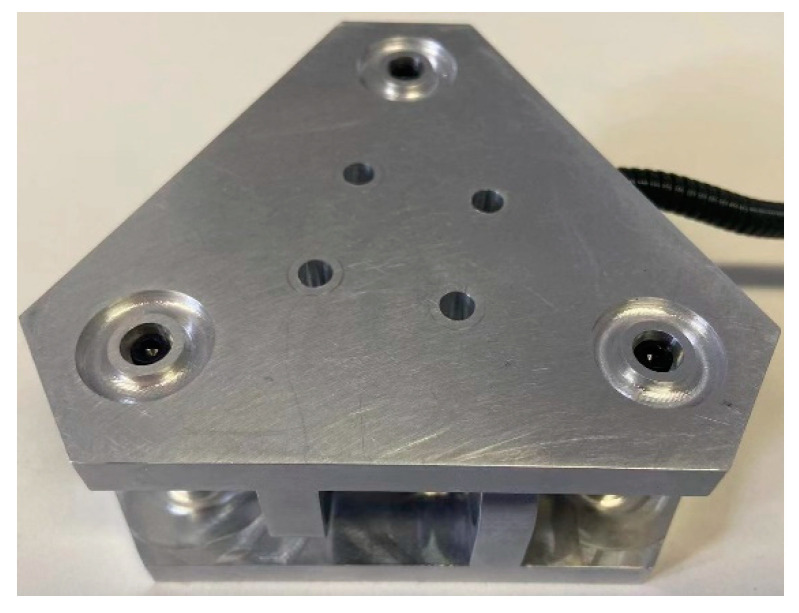
Six-dimensional force sensor model.

**Figure 13 sensors-23-07247-f013:**
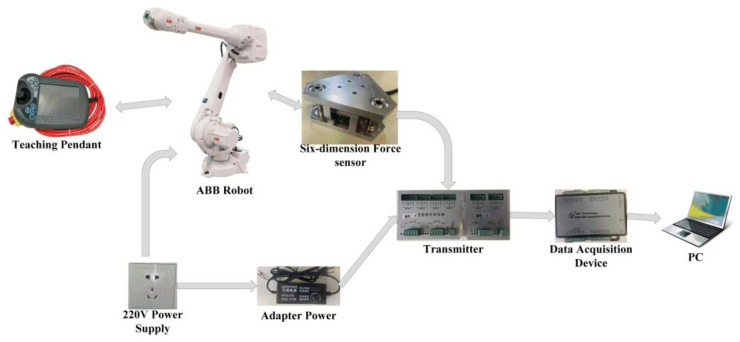
Experimental device.

**Figure 14 sensors-23-07247-f014:**
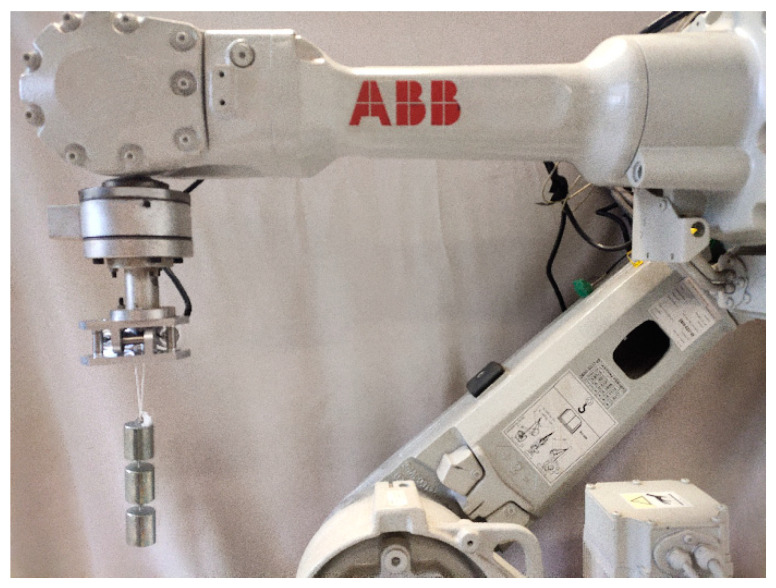
*z*-direction loading experiment.

**Table 1 sensors-23-07247-t001:** Corresponding relationship between $y_{i}$ and $\theta_{i}$.

$y_{i}$	$\theta_{i}$
0	0
$\frac{l}{4}$	$\frac{3l}{8Eb}\left(\frac{4F_{z}l\;+\;4M_{x}}{h_{1}^{3}}+\frac{3F_{z}l\;+\;4M_{x}}{h_{2}^{3}}\right)$
$\frac{l}{2}$	$\frac{3l}{4Eb}\left(\frac{2F_{z}l\;+\;2M_{x}}{h_{1}^{3}}+\frac{3F_{z}l\;+\;4M_{x}}{h_{2}^{3}}+\frac{F_{z}l\;+\;2M_{x}}{t^{3}}\right)$
$\frac{3l}{4}$	$\frac{3l}{8Eb}\left(\frac{4F_{z}l\;+\;4M_{x}}{h_{1}^{3}}+\frac{7F_{z}l\;+\;12M_{x}}{h_{2}^{3}}+\frac{4F_{z}l\;+\;8M_{x}}{t^{3}}\right)$
l	$\frac{3l\left(F_{z}l\;+\;2M_{x}\right)}{2Eb}\left(\frac{1}{h_{1}^{3}}+\frac{2}{h_{2}^{3}}+\frac{1}{t^{3}}\right)$

**Table 2 sensors-23-07247-t002:** Corresponding relationship between $y_{i}$ and $\Delta_{i}$.

$y_{i}$	$\Delta_{i}$
0	0
$\frac{l}{4}$	$\frac{3l^{2}}{64Eb}\left(\frac{4F_{z}\;+\;4M_{x}}{h_{1}^{3}}+\frac{3F_{z}\;+\;4M_{x}}{h_{2}^{3}}\right)$
$\frac{l}{2}$	$\frac{3l^{2}}{64Eb}\left(\frac{6F_{y}l\;+\;6M_{x}}{h_{1}^{3}}+\frac{6F_{y}l\;+\;8M_{x}}{h_{2}^{3}}+\frac{F_{y}l\;+\;M_{x}}{t^{3}}\right)$
$\frac{3l}{4}$	$\frac{3l^{2}}{64Eb}\left(\frac{20F_{z}l\;+\;20M_{x}}{h_{1}^{3}}+\frac{25F_{z}l\;+\;36M_{x}}{h_{2}^{3}}+\frac{8F_{z}l\;+\;16M_{x}}{t^{3}}\right)$
l	$\frac{3l^{2}}{16Eb}\left(\frac{7F_{z}l\;+\;8M_{x}}{h_{1}^{3}}+\frac{10F_{z}l\;+\;16M_{x}}{h_{2}^{3}}+\frac{4F_{z}l\;+\;8M_{x}}{t^{3}}\right)$

**Table 3 sensors-23-07247-t003:** Six-dimensional force sensor structure parameters.

E	G	a	$l_{1}$	l	$r_{1}$	$r_{2}$
197 Gpa	77 Gpa	10 mm	25 mm	50 mm	72 mm	50 mm
$r_{3}$	α	β	m	$l_{2}$	$r_{4}$	γ
50 mm	30°	90°	10 mm	35 mm	55.68 mm	21.05°

## Data Availability

Data sharing is not applicable to this article.

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
