# Peer review of "Accuracy Analysis and Experimental Study of a Six-Dimensional Force Sensor with Circular Flexible Spherical Joints"

_sensors, 2023, doi:10.3390/s23167247_

Round 1
Reviewer 1 Report
This manuscript focus on a structurally symmetric circular flexible spherical joint and applies it to a six-dimensional force sensor. The stiffness matrix of the circular flexible ball joint is derived by numerical method, and the stiffness matrix of the whole sensor is obtained, and the force mapping matrix is obtained from the stiffness matrix of the sensor. The concept of Euclidean distance was introduced to analyze the influence of the main structural parameters R, l, t on the accuracy of the sensor, and the finite element simulation analysis was carried out. The sensor prototype was developed and the experimental verification was carried out. The manuscript work has certain characteristics and innovation, and has certain reference value for readers in related fields. But there are many errors in the manuscript (not limited to the issues listed below) that need to be checked or corrected..
1. In L3, "joins" is wrong.
2. In L34, "Gauge" does not need to start with a capital letter.
3. Pls check whether the coordinates x\y in Fig. 1 are consistent with formula (1).
4. Pls check the correctness of position A5 and a5 in Fig.3.
5. “f1”-“f6” in L283-284 should be marked in Fig.3.
-
The quality of English can be improved.
Author Response
Dear Editor and Reviewer:
Thank you for your letter and for the reviewers’ comments concerning our manuscript entitled “Accuracy analysis and experimental study of a six-dimensional force sensor with circular flexible spherical joints” (ID: sensors-2548831). Those comments are all valuable and very helpful for revising and improving our paper, as well as the important guiding significance to our researches. We have studied comments carefully and have made correction which we hope meet with approval. The modified part has been marked in the paper. The main corrections in the paper and the responds to the reviewer’s comments are as flowing:
Responds to the reviewer’s comments:
1.Response to comment: (In L3, "joins" is wrong.)
Response: We have modified “joints” to “joints”.
2.Response to comment: (In L34, "Gauge" does not need to start with a capital letter.)
Response: We have modified “Guage” to start with a lowercase letter.
3.Response to comment: (Pls check whether the coordinates x\y in Fig. 1 are consistent with formula (1).)
Response: We have made modifications to the coordinates of Fig. 1, and the coordinates of Fig.1 are consistent with Formula (1).
4.Response to comment: (Pls check the correctness of position A5 and a5 in Fig.3.)
Response: We have swapped the positions and in Fig. 3.
5.Response to comment: (“f1”-“f6” in L283-284 should be marked in Fig.3)
Response: We have marked - in Fig. 3.
Special thanks to you for your good comments.
Once again, thank you very much for your comments and suggestions.
Reviewer 2 Report
The full text is reasonable, with both detailed theoretical derivation and program calculation diagrams, which is a better paper, the disadvantage is that equation (1) and equation (5) are inappropriate and need to be rewritten before publication.
The equation (1) and equation (5) are inappropriate and need to be rewritten before publication.
Author Response
Dear Editor and Reviewer:
Thank you for your letter and for the reviewers’ comments concerning our manuscript entitled “Accuracy analysis and experimental study of a six-dimensional force sensor with circular flexible spherical joints” (ID: sensors-2548831). Those comments are all valuable and very helpful for revising and improving our paper, as well as the important guiding significance to our researches. We have studied comments carefully and have made correction which we hope meet with approval. The modified part has been marked in the paper. The main corrections in the paper and the responds to the reviewer’s comments are as flowing:
Responds to the reviewer’s comments:
Response to comment: (The equation (1) and equation (5) are inappropriate and need to be rewritten before publication.)
Response: We have made modifications to equation (1) and equation (5) and marked them in the paper.
Special thanks to you for your good comments.
Once again, thank you very much for your comments and suggestions.
Reviewer 3 Report
Following your simulation and experimental results, the overall quality is good. However, couple of issues still need to be illustrated or added.
1. In Line 127, using ∂z as a variable is not a choice. Please try to use the other format to avoid the mistake as a partial derivative. So is ∂y in Line 167.
2. In Line 322, “Stiffness” should be “stiffness”. Please check the others. In Line 346, this case is repeated.
3. In Line 281, “Equation” should be “Equations”. The expression in Line 373 is not consistent.
4. In Lines 326 and 327, the grammar must be revised, especially “Where”. Please check the others.
5. In Lines 432-434, the expression of matrix is not consistent. Please revise it.
6. In Lines 435-437, if the mapping matrix is invertible, G is given as G-1. This kind of expression is not suitable. The authors must use the other matrix symbol to express it.
7. In Lines 470 and 472, “1” should be “l”.
8. In Line 575, one of matrix variables uses the wrong expression.
9. In Lines 623-635, the practical measurement is good. Please add the mean value and standard deviation during these 10-time test.
The English quality should be amended more.
Author Response
Dear Editor and Reviewer:
Thank you for your letter and for the reviewers’ comments concerning our manuscript entitled “Accuracy analysis and experimental study of a six-dimensional force sensor with circular flexible spherical joints” (ID: sensors-2548831). Those comments are all valuable and very helpful for revising and improving our paper, as well as the important guiding significance to our researches. We have studied comments carefully and have made correction which we hope meet with approval. The modified part has been marked in the paper. The main corrections in the paper and the responds to the reviewer’s comments are as flowing:
Responds to the reviewer’s comments:
1.Response to comment: (In Line 127, using ∂z as a variable is not a choice. Please try to use the other format to avoid the mistake as a partial derivative. So is ∂y in Line 167.)
Response: We have modified ∂x, ∂y, and ∂z in the text to,, and, and marked them in the paper.
2.Response to comment: (In Line 322, “Stiffness” should be “stiffness”. Please check the others. In Line 346, this case is repeated.)
Response: We have changed “Stiffness” on line 322 to “stiffness” and made modifications to the same issues in other parts of the paper.
3.Response to comment: (In Line 281, “Equation” should be “Equations”. The expression in Line 373 is not consistent.)
Response: We have modified line 322 from “Equation” to “Equations” and aligned line 373 with the other parts.
4.Response to comment: (In Lines 326 and 327, the grammar must be revised, especially “Where”. Please check the others.)
Response: We have made modifications to the grammar of lines 326 and 327, and made modifications to other parts.
5.Response to comment: (In Lines 432-434, the expression of matrix is not consistent. Please revise it.)
Response: represents the middle part of Equation (47), and represents the middle part of Equation (46). The two represent different meanings, and for Equation (48), it is the abbreviation of Equation (47). Its meaning has been explained in the paper.
6.Response to comment: (In Lines 435-437, if the mapping matrix is invertible, G is given as G-1. This kind of expression is not suitable. The authors must use the other matrix symbol to express it.)
Response: We have changed the expression in the paper to “The force mapping matrix G of the sensor is represented as follows.”
7.Response to comment: (In Lines 470 and 472, “1” should be “l”.)
Response: We have modified the “1” of lines 470 and 472 to “l”.
8.Response to comment: (In Lines 432-434, the expression of matrix is not consistent. Please revise it.)
Response: We have modified line 575 fromto.
9.Response to comment: (In Lines 432-434, the expression of matrix is not consistent. Please revise it.)
Response: In order to verify the good measurement results, the mean value and standard deviation were introduced during the experimental process. By using the mean value and standard deviation, it can be determined that different loading values result in good experimental results. The specific expression has been provided in the paper.
Special thanks to you for your good comments.
Once again, thank you very much for your comments and suggestions.